# HSELKD: SELECTIVE KNOWEDGE DISTILLATION FOR HYPERGRAPHS USING OPTIMAL TRANSPORT

## ABSTRACT

Hypergraph Neural Networks (HGNNs) excel at modeling high-order dependencies through hyperedges, but their heavy inference cost limits deployment in latency-sensitive industrial scenarios. Knowledge Distillation (KD) offers a promising way to combine the expressiveness of graph-based models with the efficiency of lightweight Multi-Layer Perceptrons (MLPs). However, existing KD methods typically transfer the full output distribution of the teacher, overlooking the practical setting where only a subset of knowledge is necessary or beneficial. To address this, we propose HSelKD, a selective KD framework that transfers task-relevant knowledge from an HGNN teacher to a lightweight MLP student. HSelKD leverages Inverse Optimal Transport to distill the most informative parts of the teacher's knowledge in a capacity-aware manner. We further introduce two principled variants: (1) Task-Aware Distillation, which specializes the student on task-relevant labels, and (2) Reject-Aware Distillation, which equips the student with the ability to abstain from uncertain or out-of-scope predictions. Extensive experiments on hypergraph and graph benchmarks show that HSelKD consistently outperforms lightweight baselines, matches the accuracy of structure-aware teachers, and delivers faster inference by up to $53\times$ with lower training cost and computational overhead. These results establish HSelKD as a practical and scalable solution for real-world, latency-constrained deployments.

## 1 INTRODUCTION

HGNNs have emerged as a powerful paradigm for capturing higher-order relationships beyond traditional pairwise graphs, enabling richer representations in domains such as citation networks, social interactions, and recommendation systems Kim et al. (2024). Despite their expressive power, HGNNs face fundamental practical limitations. In particular, deep HGNN architectures suffer from explosive inference costs. Because message passing involves both nodes and hyperedges, the number of interactions grows rapidly with network depth. Even moderate-sized hyperedges can make deep HGNNs prohibitively costly, as illustrated in Figure 1a, where the number of feature fetches escalates steeply with additional layers. This inefficiency renders even accurate HGNNs impractical for large-scale or latency-sensitive applications.

**Motivation**. To address this challenge, KD offers a promising solution, where a large, expensive teacher model transfers its knowledge to a smaller student model, enabling significant reductions in inference latency while preserving predictive performance Zhang et al. (2021); Feng et al. (2024); Forouzandeh et al. (2025). However, most KD pipelines transfer the full output distribution of the teacher, implicitly assuming that all knowledge is equally relevant. This often introduces redundant or irrelevant knowledge, which is especially harmful in subtask-specific or open-world scenarios Shi (2025). Moreover, when only a subset of labels is needed, conventional pipelines require retraining the teacher on that subset, which is an impractical burden in large-scale settings. In practice, downstream tasks frequently require only a subset of the teacher's knowledge, underscoring the need for selective, task-adaptive distillation.

**Present work and novelties**. To overcome these limitations, we propose HSelKD, a hypergraph-selective knowledge distillation framework grounded in optimal transport (OT). Our method aligns teacher and student distributions through OT (implemented via Sinkhorn iterations), incorporates instance-level contrastive alignment, and enforces high-order consistency between node- and

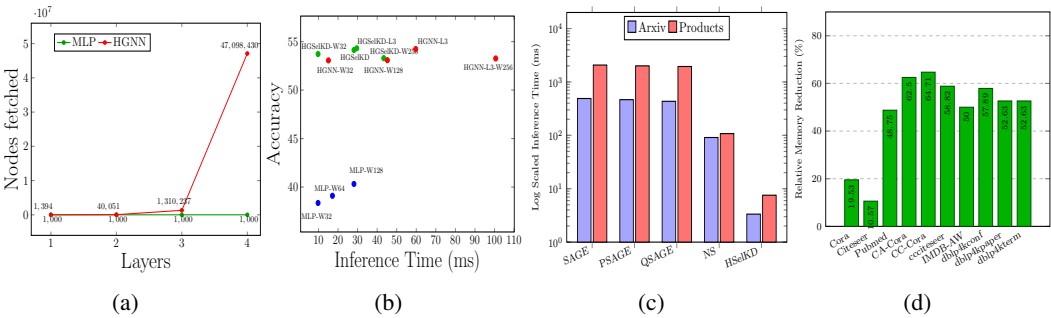

Figure 1: (a) Number of Nodes Fetched on CA-Cora, (b) Inference time vs Accuracy (c) Inference time (in log-scale milliseconds) on Arxiv and Products. (d) Relative memory reduction.

hyperedge-level predictions. On top of this foundation, we introduce two extensions: (i) *Task-Aware HSelKD*, which transfers only task-relevant knowledge to reduce negative transfer; and (ii) *Reject-Aware HSelKD*, which introduces a reject channel to handle uncertain or out-of-distribution predictions (i.e., samples that deviate from the training data distribution), improving robustness under restricted or imbalanced label spaces. Our contributions can be summarized as follows:

- We cast hypergraph distillation as a bi-level OT optimization problem that couples student predictions with the teacher while enforcing structural consistency. Unlike prior works such as LightHGNN$^+$, which rely on heuristic edge reliability, HSelKD offers a principled OT-based formulation that additionally supports selective knowledge transfer.

- We design task-aware and reject-aware modes, enabling the student to absorb only relevant knowledge and explicitly handle uncertain or out-of-distribution predictions, enhancing both efficiency and robustness.

- By integrating OT-based KL distillation with contrastive alignment and hypergraph constraints, the student retains the representational power of HGNNs while achieving the efficiency of a simple MLP.

- Extensive experiments on several graph and hypergraph benchmarks, across transductive and inductive settings, show that HSelKD consistently outperforms lightweight baselines such as LightHGNN$^+$, achieves accuracy competitive with structure-aware HGNNs, and substantially improves inference efficiency.

## 2 BACKGROUND AND RELATED WORKS

Full details of the hypergraph neural networks (HGNNs), optimal transport (OT/IOT), and related works are provided in Appendix A.

## 3 METHODOLOGY

**Problem Formulation.** We consider a hypergraph $\mathcal{H} = (V, \mathcal{E}, X, Y, H)$, where $V$ denotes the set of nodes with $|V| = N$, $\mathcal{E} = \{e_1, \ldots, e_M\}$ is the set of hyperedges, $X \in \mathbb{R}^{N \times d}$ is the node feature matrix with $d$-dimensional attributes, and $Y = \{y_v \mid v \in V\}$ denotes the node labels from $K$ classes. The structural information of the hypergraph is captured by the incidence matrix $H \in \{0, 1\}^{N \times M}$, where $H(v, e) = 1$ indicates that node $v$ participates in hyperedge $e$. The node set is partitioned into a labeled subset $D_L = (V_L, Y_L)$ and an unlabeled subset $D_U = (V_U, Y_U)$ with $V_U = V \setminus V_L$. The teacher model, a hypergraph neural network (HGNN) $f_T$, is trained on $D_L$ to capture high-order dependencies across all classes. The objective is then to distill into a lightweight student model $f_S$, instantiated as an multi-layer perceptron (MLP), only the knowledge relevant to a target subset of classes $\mathcal{C}_{sel} \subseteq \{1, \ldots, K\}$. This selective transfer avoids retraining the teacher and reduces computational overhead while ensuring efficient and task-relevant knowledge distillation.

We propose HGSelKD, a novel selective distillation framework for node classification on hypergraphs, designed to transfer task-relevant knowledge from a high-capacity hypergraph neural net-

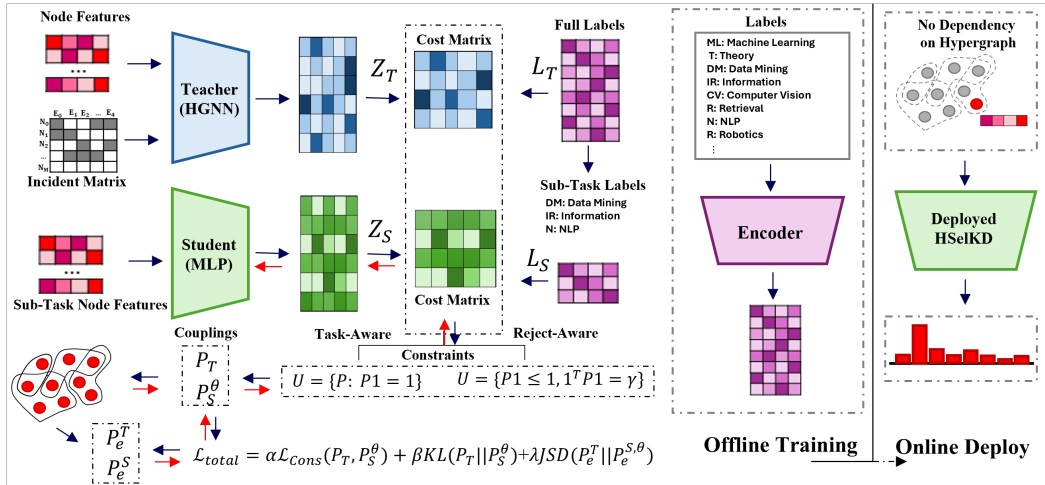

Figure 2: Architecture of the proposed HSelKD. The teacher HGNN provides high-order supervision via OT-based couplings $P_T$, while the student MLP generates $P_S^\theta$. Task-aware and reject-aware constraints regulate the couplings for different deployment scenarios. After training, the student model operates graph-free, enabling lightweight and efficient inference

work (HGNN) teacher to a lightweight, graph-free student (MLP). In contrast to standard distillation that imitates the teacher's full output distribution, HSelKD leverages an OT-driven selective transfer mechanism. By aligning the student solely with task-relevant classes and their hyperedge-induced higher-order semantics, our approach captures richer structural dependencies while avoiding redundant or task-irrelevant knowledge. The HGSElKD supports two deployment modes: Target-Aware mode, which distills knowledge for a predefined class subset, and Reject-Aware mode, which augments the label space with a reject class to handle out-of-scope inputs, thereby improving robustness in open-world scenarios. Figure 2 illustrates the overall architecture, emphasizing the selective, high-order knowledge transfer process between teacher and student.

## 3.1 BI-LEVEL INVERSE OT-BASED DISTILLATION

We extend inverse OT-based distillation into a bi-level formulation tailored for knowledge transfer in hypergraphs. The key idea is to align the student's transport plan with the teacher's, not by directly imitating logits, but by matching their induced coupling matrices, which capture structured relationships between nodes and class-level semantics. The outer level updates the student parameters $\theta$ by aligning its transport plan with that of the teacher, while the inner level solves two OT problems to compute the teacher and student transport plans.

**Inner Level (OT subproblems).** Given a pre-trained teacher HGNN $f_T$ and a student MLP $f_S$, each node $v \in V$ is mapped to a representation $z_T = f_T(x_v)$ or $z_S = f_S(x_v)$. Each class $k \in \{1, \dots, K\}$ is represented by a prototype $L = g(y_k) \in \mathbb{R}^K$, instantiated as a one-hot basis vector. The cost matrices are then defined as the negative similarity between node representations and class prototypes: $(C_T)_{ij} = -z_T(x_i) \cdot g(y_j)$, $(C_S^\theta)_{ij} = -z_S(x_i) \cdot g(y_j)$. The corresponding transport plans are obtained by solving entropic OT problems:

$$P_T = \arg \min_{P \in \Pi(\mu,\nu)} \langle C_T, P \rangle - \varepsilon H(P), \qquad P_S^\theta = \arg \min_{P \in \Pi(\mu,\nu)} \langle C_S^\theta, P \rangle - \varepsilon H(P),$$

where the node marginal $\mu$ is a uniform distribution over nodes $\mu_i = 1/N$, the class marginal $\nu$ is a uniform distribution over the selected label subset $C_{sel}(\nu_j = 1/|C_{sel}|)$, and $\Pi(\mu,\nu) = \{P \in \mathbb{R}_{\geq 0}^{n \times K} \mid P\mathbf{1} = \mu, \ P^\top \mathbf{1} = \nu\}$ denotes the set of admissible couplings with prescribed marginals $\mu$ (over nodes) and $\nu$ (over classes), and $H(P)$ is the entropy regularizer. These subproblems yield couplings that describe how node representations align with class prototypes under both teacher and student models.

**Outer Level (student update).** Given the teacher and student transport plans $P_T$ and $P_S^\theta$, we update the student parameters $\theta$ by solving:

$$\min_\theta \; \alpha \, \mathcal{L}_{\text{con}}(P_T, P_S^\theta) \; + \; \beta \, \text{KL}(P_T \parallel P_S^\theta).$$

The contrastive loss $\mathcal{L}_{\text{con}}$ preserves structural similarity induced by the teacher, while the KL term enforces distributional alignment between couplings. Importantly, $\mathcal{L}_{\text{con}}$ enables unsupervised student training, since it depends only on the teacher's induced structure rather than ground-truth labels.

**Hyperedge-aware topology-enhanced KD.** To fully exploit the higher-order dependencies encoded in hypergraphs, we extend the bi-level OT framework with a hyperedge-aware objective. The key idea is that, beyond aligning node-to-class couplings, the student should also preserve structural consistency at the *hyperedge level*, where groups of nodes jointly contribute to semantic relations. Formally, let $p_v^S$ and $p_v^T$ denote the OT-based couplings (at temperature $\tau$) for the student and teacher at node $v$. We aggregate these couplings to the hyperedge level by averaging over the incident nodes of each hyperedge $e$ with degree $d_e$: $p_e^S = \frac{1}{d_e} \sum_{v \in e} p_v^S,$ $\quad p_e^T = \frac{1}{d_e} \sum_{v \in e} p_v^T$. These aggregated distributions encode *hyperedge-level couplings*, which capture how sets of nodes collectively align with class prototypes. To enforce alignment at this level, we apply the Jensen–Shannon divergence (JSD) Lin (2002), a symmetric and smoothed variant of KL divergence commonly used in distillation Cui et al. (2024): $\text{JSD}(p_e^T \parallel p_e^S) = \frac{1}{2} \text{KL}(p_e^T \parallel m_e) + \frac{1}{2} \text{KL}(p_e^S \parallel m_e), \quad m_e = \frac{1}{2}(p_e^T + p_e^S)$. The final HSelKD objective integrates three complementary terms:

$$\mathcal{L}_{\text{total}} = \alpha \, \mathcal{L}_{\text{con}}(P_T, P_S^\theta) + \beta \, \text{KL}(P_T \parallel P_S^\theta) + \lambda \, \text{JSD}(p_e^T \parallel p_e^S),$$

where $\alpha, \beta, \gamma$ balance contrastive alignment, node-level coupling consistency, and hyperedge-level distribution matching. This ensures that the student not only mimics the teacher at the node level, but also inherits the higher-order structural semantics encoded in hyperedges.

## 3.2 Selective Distillation via Inverse OT

To support different deployment scenarios, HSelKD includes two modes: (1) *Task-Aware HSelKD*, which trains the student on a restricted subset of classes to enhance specialization, and (2) *Reject-Aware HSelKD*, which extends this by incorporating a reject class to handle out-of-scope inputs, improving robustness in open-world scenarios. The pseudocode for HSelKD covering both the task-aware and reject-aware modes is provided in the appendix (Algorithms 1 and 2) together with reproducibility details.

**Task-Aware HSelKD.** Formally, given a class subset $\mathcal{C}_{\text{sel}} \subseteq \{1, \ldots, K\}$ and a batch of node representations $\{f_T(x_i), f_S(x_i)\}_{i=1}^N$, we restrict the distillation to this subset by constructing a selective cost matrix $\tilde{C}_T \in \mathbb{R}^{N \times |\mathcal{C}_{\text{sel}}|}$ based on the similarity between nodes and class prototypes $g(y_j), y_j \in \mathcal{C}_{\text{sel}}$. The corresponding teacher and student transport plans are obtained via entropic OT as $\tilde{P}_T = \arg\min_{P \in \tilde{\mathcal{U}}} \langle \tilde{C}_T, P \rangle - \epsilon H(P)$, $\tilde{P}_S^\theta = \arg\min_{P \in \tilde{\mathcal{U}}} \langle \tilde{C}_S^\theta, P \rangle - \epsilon H(P)$. The outer-level distillation objective then minimizes

$$\min_\theta \; \alpha \, \mathcal{L}_{\text{con}}(\tilde{P}_T, \tilde{P}_S^\theta) + \beta \, \text{KL}(\tilde{P}_T \parallel \tilde{P}_S^\theta) + \lambda \, \text{JSD}(\tilde{p}_e^{T,\theta} \parallel \tilde{p}_e^S),$$

where $\tilde{\mathcal{U}}$ denotes the admissible couplings constrained to $\mathcal{C}_{\text{sel}}$ and $H(P)$ is the entropy regularizer. This inverse OT formulation enables task-aware supervision, since both cost matrices and coupling constraints are restricted to $\mathcal{C}_{\text{sel}}$.

**Reject-Aware HSelKD.** To support scenarios where the student may face categories outside its target label set, we extend the objective with a reject class under unbalanced partial OT constraints:

$$\min_\theta \quad \alpha \, \mathcal{L}_{\text{con}}(\tilde{P}_T, \tilde{P}_S^\theta) + \beta \, \text{KL}(\tilde{P}_T \parallel \tilde{P}_S^\theta) + \lambda \, \text{JSD}(\tilde{p}_e^T \parallel \tilde{p}_e^{S,\theta}),$$

$$\text{s.t.} \quad \tilde{P}_T = \arg\min_{\tilde{P}\mathbf{1} \leq \mathbf{1}, \, \mathbf{1}^\top \tilde{P}\mathbf{1} = \gamma} \langle \tilde{C}_T, \tilde{P} \rangle - \varepsilon H(\tilde{P}), \quad (1)$$

$$\tilde{P}_S^\theta = \arg\min_{\tilde{P}\mathbf{1} \leq \mathbf{1}, \, \mathbf{1}^\top \tilde{P}\mathbf{1} = \gamma} \langle \tilde{C}_S^\theta, \tilde{P} \rangle - \varepsilon H(\tilde{P}).$$

where $\gamma$ denotes the number of samples belonging to $\mathcal{C}_{\text{sel}}$. Here, $\tilde{P}_T$ and $\tilde{P}_S^\theta$ are selective transport plans computed over the extended label set $\mathcal{C}_{\text{sel+}} = \mathcal{C}_{\text{sel}} \cup \{n+1\}$, where $n+1$ denotes a reject class aggregating all out-of-subset categories.

### 3.3 THEORETICAL ANALYSIS

To better understand the foundations of HSelKD, we provide theoretical guarantees that clarify its optimization properties and the soundness of its selective design. We focus on three aspects: (i) differentiability of the training objective, (ii) consistency of OT-based alignment, and (iii) guarantees for the task-aware, reject-aware, and selective distillation modes. Formal proofs for all results are deferred to Appendix E.

**Lemma 1** (Differentiability of the Objective). *Let $P_S^\theta$ denote the student's entropic OT plan with regularization $\varepsilon > 0$. If the student cost matrix $C_S^\theta$ is differentiable in $\theta$, then the full HSelKD objective is differentiable in $\theta$. The proof is provided in Appendix E.*

**Lemma 2** (Consistency of OT Alignment). *Assume the student model has sufficient capacity. Minimizing $KL(P_S^\theta \| P_T)$ yields $P_S^\theta \to P_T$ at convergence. The proof is provided in Appendix E.*

**Theorem 1** (Task-Aware Optimality). *Let $\mathcal{C}_{sel} \subseteq \mathcal{C}$ be a subtask-specific label set. If the teacher distributions $P_T$ are supported only on $\mathcal{C}_{sel}$, then the optimal restricted student plan $\widetilde{P}_S^\theta$ recovers the projection of teacher knowledge onto $\mathcal{C}_{sel}$. The proof is provided in Appendix E.*

**Theorem 2** (Reject-Aware Robustness). *Consider inputs $x$ outside the target label set $\mathcal{C}_{sel}$. If $C_S^\theta(x, r) < C_S^\theta(x, j)$ for all $j \in \mathcal{C}_{sel}$, where $r$ denotes the reject class, then under partial OT constraints, the optimal student plan satisfies $P_S(x, r) \to 1$. The proof is provided in Appendix E.*

**Theorem 3** (Selective Distillation Soundness). *Let $\mathcal{C}_{sel} \subseteq \mathcal{C}$ be the selected label subset for a subtask. Restricting the student plan $P_S^\theta$ to $\mathcal{C}_{sel}$ reduces the risk of negative transfer, i.e.,*

$$\min_\theta KL(\widetilde{P}_S^\theta \| \widetilde{P}_T) \leq \min_\theta KL(P_S^\theta \| P_T),$$

*where $\widetilde{P}$ denotes the projection of a transport plan onto $\mathcal{C}_{sel}$. The proof is provided in Appendix E.*

## 4 EXPERIMENTS

We evaluate our approach on *graph benchmarks* including Cora Sen et al. (2008), Citeseer Giles et al. (1998), and Pubmed McCallum et al. (2000), as well as *hypergraph datasets* including CA-Cora, CC-Cora, and CC-Citeseer Yadati et al. (2019), IMDB-AW Fu et al. (2020), DBLP-Paper, DBLP-Term, and DBLP-Conf Sun et al. (2011), along with the large-scale datasets Recipe-100k and Recipe-200k, and the OGB benchmarks Arxiv and Products Hu et al. (2021). We compare our framework against six representative categories of baselines. *MLP* Taud & Mas (2017), a non-graph baseline, classical *graph neural networks*, including GCN Kipf (2016) and GAT Velickovic et al. (2017) *Hypergraph neural networks*, such as HGNN Feng et al. (2019) and HGNN+ Gao et al. (2022a), *GNN-to-MLP* distillation methods, including GLNN Zhang et al. (2021), KRD Wu et al. (2023), and NOSMOG Tian et al. (2022), *HGNN-to-MLP* approaches, such as LightHGNN and LightHGNN$^+$ Feng et al. (2024), and a *HGNN-to-LightGNN* method, DistillHGNN Forouzandeh et al. (2025).Experiments are evaluated in two setups: (i) a standard transductive node classification setting, and (ii) a production setting that combines transductive and inductive predictions to better reflect real-world deployment. All results are averaged over 10 random seeds, and we report mean accuracy with standard deviation.Additional details provided in Appendix B.

**Comparing under transductive and inductive settings.** We evaluate the performance of HSelKD in a transductive scenario, focusing on full-task classification across seven hypergraph datasets (Table 1). Accuracy scores are reported along with absolute and relative improvements ($\Delta$) over each baseline. HSelKD consistently achieves higher accuracy than LightHGNN$^+$, demonstrating the effectiveness of the OT-based distillation strategy. On average, HSelKD outperforms MLP by 16.26 points (28.7%), HGNN by 1.15 points (1.7%), and LightHGNN$^+$ by 1.60 points (2.4%). Compared with LightHGNN$^+$, HSelKD benefits from more effective transfer of structural knowledge from the HGNN teacher, leading to superior predictive performance on nearly all datasets.

To evaluate HSelKD in more realistic scenarios, we adopt the production setting, which encompasses both transductive and inductive prediction across seven hypergraph datasets (Table 2). Consistent with earlier findings, HSelKD delivers strong improvements over baseline models. On average, it surpasses MLP by 14.58 points, outperforms LightHGNN$^+$ by 1.33 points, and achieves performance comparable to HGNN (–0.38 on average). These gains demonstrate that HSelKD successfully balances scalability with accuracy by leveraging knowledge transfer from a stronger teacher.

Table 1: Transductive results with absolute and relative accuracy gains ($\Delta$).

| Dataset | MLP | HGNN | LightHGNN$^+$ | HSelKD | $\Delta_{\text{MLP}}$ | $\Delta_{\text{HGNN}}$ | $\Delta_{\text{LightHGNN}^+}$ |
|---|---|---|---|---|---|---|---|
| *CA-Cora* | 63.53±2.70 | 72.14±2.54 | 73.77±1.55 | **75.42±2.02** | 11.89 (18.7%) | 3.28 (4.5%) | 1.65 (2.2%) |
| *CC-Cora* | 51.86±1.47 | 68.97±3.12 | 69.53±2.96 | **71.48±2.37** | 19.62 (37.8%) | 2.51 (3.6%) | 1.95 (2.8%) |
| *CC-Citeseer* | 51.79±2.59 | 63.66±1.17 | 64.77±1.69 | **67.45±1.57** | 15.66 (30.2%) | 3.79 (6.0%) | 2.68 (4.1%) |
| *DBLP-Paper* | 62.84±1.58 | 71.74±0.90 | 71.71±1.70 | **73.16±1.36** | 10.32 (16.4%) | 1.42 (2.0%) | 1.45 (2.0%) |
| *DBLP-Term* | 62.84±1.58 | **82.40±1.71** | 79.99±1.34 | 81.89±1.64 | 19.05 (30.3%) | -0.51 (-0.6%) | 1.90 (2.4%) |
| *DBLP-Conf* | 62.84±1.58 | **94.01±0.21** | 90.38±0.69 | 90.45±0.44 | 27.61 (44.0%) | -3.56 (-3.8%) | 0.07 (0.1%) |
| *IMDB-AW* | 40.87±1.43 | 50.42±1.70 | 49.09±3.75 | **50.54±2.74** | 9.67 (23.7%) | 0.12 (0.2%) | 1.45 (3.0%) |
| **Avg. Rank/Avg.** | 4.0 | 2.14 | 2.57 | **1.28** | 16.26 (28.7%) | 1.15 (1.7%) | 1.60 (2.4%) |

Table 2: Experimental results under production setting.

| Dataset | Setting | MLP | HGNN | LightHGNN$^+$ | HSelKD | $\Delta_{\text{MLP}}$ | $\Delta_{\text{HGNN}}$ | $\Delta_{\text{LightHGNN}^+}$ |
|---|---|---|---|---|---|---|---|---|
| | Prod. | 50.73±1.43 | 70.73±2.84 | 71.68±1.48 | 73.35±2.66 | 22.62 (44.6%) | 2.62 (3.7%) | 1.67 (2.3%) |
| CA-Cora | Tran. | 50.75±1.64 | 70.90±0.00 | 73.35±1.85 | 74.43±2.65 | 23.68 (46.6%) | 3.53 (5.0%) | 1.08 (1.5%) |
| | Ind. | 50.67±1.44 | 69.75±0.00 | 64.98±1.65 | 69.00±2.96 | 18.33 (36.2%) | -0.75 (-1.1%) | 4.02 (6.2%) |
| | Prod. | 54.41±1.36 | 68.57±3.51 | 68.61±3.26 | 70.08±2.49 | 15.67 (28.8%) | 1.51 (2.2%) | 1.47 (2.1%) |
| CC-Cora | Tran. | 54.42±1.52 | 68.49±3.46 | 69.52±3.07 | 70.42±1.98 | 15.99 (29.4%) | 1.93 (2.8%) | 0.90 (1.3%) |
| | Ind. | 54.36±1.14 | 66.75±3.98 | 64.98±4.13 | 68.74±4.95 | 14.38 (26.5%) | 1.99 (3.0%) | 3.76 (5.8%) |
| | Prod. | 54.41±1.36 | 64.09±1.20 | 64.13±1.37 | 66.26±2.13 | 11.85 (21.8%) | 2.17 (3.4%) | 2.13 (3.3%) |
| CC-Citeseer | Tran. | 54.42±1.52 | 63.74±1.13 | 64.43±1.32 | 66.75±2.31 | 12.33 (22.7%) | 3.01 (4.7%) | 2.32 (3.6%) |
| | Ind. | 54.36±1.14 | 63.28±1.87 | 62.93±2.49 | 64.32±1.59 | 9.96 (18.3%) | 1.04 (1.6%) | 1.39 (2.2%) |
| | Prod. | 63.23±1.48 | 71.93±0.72 | 72.23±0.52 | 72.66±1.16 | 9.43 (14.9%) | 0.73 (1.0%) | 0.43 (0.6%) |
| DBLP-Paper | Tran. | 62.97±1.69 | 71.11±1.05 | 71.90±0.74 | 72.32±1.43 | 9.35 (14.8%) | 1.21 (1.7%) | 0.42 (0.6%) |
| | Ind. | 64.25±1.75 | 73.06±1.99 | 73.48±1.68 | 74.01±1.49 | 9.76 (15.2%) | 0.95 (1.3%) | 0.53 (0.7%) |
| | Prod. | 63.56±1.15 | 82.00±1.98 | 79.89±1.57 | 81.11±0.00 | 17.55 (27.6%) | -0.89 (-1.1%) | 1.22 (1.5%) |
| DBLP-Term | Tran. | 63.37±1.17 | 82.02±2.29 | 80.02±1.86 | 81.17±1.79 | 17.80 (28.1%) | -0.85 (-1.0%) | 1.15 (1.4%) |
| | Ind. | 64.30±1.50 | 82.90±1.98 | 79.41±1.66 | 80.89±1.34 | 16.59 (25.8%) | -2.01 (-2.4%) | 1.48 (1.9%) |
| | Prod. | 63.56±1.15 | 94.01±0.17 | 88.97±0.51 | 89.11±0.29 | 25.55 (40.2%) | -4.90 (-5.2%) | 0.14 (0.2%) |
| DBLP-Conf | Tran. | 63.37±1.17 | 93.33±0.24 | 90.67±0.72 | 90.67±0.54 | 27.29 (43.1%) | -2.67 (-2.9%) | 0.00 (0.0%) |
| | Ind. | 64.30±1.50 | 94.23±0.49 | 82.15±1.04 | 82.90±0.71 | 18.60 (28.9%) | -11.33 (-12.0%) | 0.75 (0.9%) |
| | Prod. | 47.03±1.83 | 50.93±1.06 | 48.94±1.43 | 50.19±2.23 | 3.16 (6.7%) | -0.74 (-1.5%) | 1.25 (2.6%) |
| IMDB-AW | Tran. | 45.05±3.22 | 50.39±0.85 | 49.41±1.42 | 50.88±2.16 | 5.83 (12.9%) | 0.49 (1.0%) | 1.47 (3.0%) |
| | Ind. | 47.03±1.99 | 52.38±2.81 | 47.07±2.54 | 47.43±3.20 | 0.40 (0.9%) | -4.95 (-9.5%) | 0.36 (0.8%) |
| Avg. Rank/Avg. | – | 4.00 | 2.00 | 2.57 | 1.42 | 14.58 | -0.38 | 1.33 |

**Effect of Full-Scope vs. Matched-Scope Teacher.** To investigate the influence of the teacher's knowledge scope, we consider two strategies: distilling knowledge from (i) a Full-Scope Teacher, trained on the entire label space, and (ii) a Matched-Scope Teacher, trained only on the student's restricted label set. The subtasks cover progressively larger label sets, including 2, 3, 4, and 5 labels. As shown in Table 3, the Full-Scope Teacher consistently delivers stronger results with higher Accuracy on nearly all subtasks. For instance, on CA-Cora we observe +0.99 average Accuracy over the Matched-Scope counterpart, while on CC-Cora the margin increases to +1.57 Accuracy. Results for Macro-F1 and Micro-F1 are provided in Appendix F.2.

Table 3: Accuracy of Full-Scope vs. Matched-Scope teachers on 2–5 label subtasks.

| Dataset | Type | Task1 | Task2 | Task3 | Task4 | Avgerage |
|---|---|---|---|---|---|---|
| CA-Cora | Full-Task | **95.14±0.77** | **88.86±2.02** | **81.53±2.28** | **79.15±3.40** | **86.17** |
| | Sub-Task | 94.65±0.79 | 87.84±1.58 | 81.10±2.38 | 79.13±2.08 | 85.18 |
| CC-Cora | Full-Task | **93.78±1.30** | 86.01±1.57 | **79.11±1.96** | **77.43±3.39** | **84.08** |
| | Sub-Task | 92.65±1.15 | **86.01±1.15** | 77.71±3.31 | 75.65±3.11 | 82.51 |
| CC-Citeseer | Full-Task | **92.48±1.22** | **85.51±1.61** | **79.33±1.18** | 71.12±3.15 | **82.11** |
| | Sub-Task | 91.78±1.07 | 83.81±1.51 | 77.83±1.70 | **71.60±0.76** | 81.26 |

**Comparing HSelKD with LightHGNN$^+$ across Subtasks.** We compare HSelKD with the full-distribution baseline LightHGNN$^+$ across CC-Citeseer subtasks (2–5 labels). LightHGNN$^+$ is re-trained independently for each subtask, whereas HSelKD reuses a single teacher trained on the full label space across tasks. As shown in Table 4, HSelKD consistently outperforms LightHGNN$^+$ in terms of Accuracy, with relative gains ranging from 0.91% to 4.81% and an average improvement of 2.57%. Corresponding results for Micro-F1 and Macro-F1 are reported in Appendix F.3.

Table 4: Accuracy comparison between HSelKD and LightHGNN$^+$ on CC-Citeseer subtasks.

| Model | Task1 | Task2 | Task3 | Task4 | Average |
|---|---|---|---|---|---|
| HSelKD | **92.48±1.22** | **85.51±1.61** | **79.33±1.18** | **71.12±3.15** | **82.11** |
| LightHGNN | 91.65±1.28 | 83.16±1.75 | 75.69±1.53 | 69.72±1.25 | 80.30 |
| delta | 0.83 (0.91%) | 2.35 (2.83%) | 3.64 (4.81%) | 1.40 (2.01%) | 2.06 (2.57%) |

**Selective IOT-based KD vs. Standard KD.** A key component of HSelKD is its selective distillation mechanism, which employs Optimal Transport to align and transfer only the logits corresponding to each subtask's label set. To assess its effectiveness, we adapt vanilla KD to a partial-label setting, where the student is trained on subtask-specific classes while the teacher covers the full label space. Table 5 compares HSelKD with extended KD under identical training conditions. HSelKD consistently improves accuracy. On CA-Cora and CC-Cora, the gains are modest (+0.4 to +1.8 points), while on CC-Citeseer they are more substantial (up to +4.1 on Task 3).

Table 5: Performance of HSelKD and the extended vanilla KD method with HGNN as teacher.

| Task | CA-Cora | | | CC-Cora | | | CC-Citeseer | | |
|---|---|---|---|---|---|---|---|---|---|
| | HSelKD | KD | delta | HSelKD | KD | delta | HSelKD | KD | delta |
| Task1 | **95.1±0.8** | 93.7±1.0 | +1.5 (1.6%) | **93.8±1.3** | 92.0±1.8 | +1.8 (1.9%) | 92.5±1.2 | 91.8±1.9 | +0.7 (0.8%) |
| Task2 | 88.9±2.0 | 87.9±1.4 | +1.0 (1.1%) | 86.0±1.6 | 85.5±1.3 | +0.6 (0.7%) | 85.5±1.6 | 83.4±1.4 | +2.1 (2.5%) |
| Task3 | 81.5±2.3 | 80.5±2.4 | +1.1 (1.3%) | 79.1±2.0 | 78.7±3.1 | +0.4 (0.5%) | 79.3±1.2 | 75.2±1.5 | **+4.1 (5.5%)** |
| Task4 | 79.2±3.4 | 78.3±2.7 | +0.9 (1.2%) | 77.4±3.4 | 75.6±3.5 | **+1.8 (2.4%)** | 71.1±3.2 | 69.6±0.7 | +1.5 (2.2%) |
| Avg. | **86.17** | 85.07 | +1.1 (1.3%) | **84.08** | 82.95 | +1.1 (1.4%) | **82.11** | 79.99 | +2.1 (2.2%) |

**Reject-Aware Evaluation.** We evaluate HSelKD in reject-aware scenarios, where the student must classify in-scope classes while rejecting out-of-scope inputs. Each subtask is extended with an additional reject class, yielding four settings with 2–5 in-scope labels plus rejection. As shown in Table 6, HSelKD consistently outperforms standard KD across CA-Cora and CC-Citeseer. On average, it achieves +6.51 Accuracy on CA-Cora and +5.60 Accuracy on CC-Citeseer, with the largest gain of +13.23 Accuracy observed on CA-Cora Task 3. These results confirm that explicit rejection of off-target mass sharpens decision boundaries and improves robustness under restricted label spaces. Corresponding Micro-F1 and Macro-F1 results are provided in Appendix F.4.

Table 6: Reject aware Comparison of HSelKD vs the extended vanilla KD method on four subtasks.

| Dataset | Type | Task1 | Task2 | Task3 | Task4 | Avg |
|---|---|---|---|---|---|---|
| CA-Cora | HSelKD | 85.29±1.51 | 84.70±1.31 | 75.71±2.49 | 74.34±2.49 | **80.01** |
| | KD | 82.53±5.62 | 81.56±6.18 | 62.48±1.26 | 69.43±6.18 | 73.50 |
| | delta | +2.76 | +3.14 | +13.23 | +4.91 | +6.51 |
| CC-Citeseer | HSelKD | 81.04±1.27 | 75.91±0.90 | 66.62±1.67 | 67.11±1.44 | **72.17** |
| | KD | 73.48±5.61 | 67.67±6.71 | 62.04±4.34 | 63.07±2.65 | 66.57 |
| | delta | +7.56 | +8.24 | +4.58 | +4.04 | +5.60 |

**Comparison with Hypergraph and Distillation Baselines.** Table 7 summarizes results on three hypergraph benchmarks (CA-Cora, DBLP-Paper, IMDB-AW). HSelKD achieves the best accuracy on CA-Cora and DBLP-Paper, slightly surpassing HGNN$^+$ and DistillHGNN, while on IMDB-AW performance remains competitive with HGNN$^+$. Comparisons on graph datasets (Cora, Pubmed, Citeseer) and additional baselines are provided in Appendix F.5.

**Accuracy–Efficiency Trade-offs of HSelKD.** Figures 1(b–d) evaluate HSelKD from three complementary perspectives. In Figure 1(b), we examine the trade-off between accuracy and inference time. HSelKD maintains accuracy comparable to or better than HGNN variants while achieving much lower latency. (Here, $L$ denotes the number of layers and $W$ the hidden dimension size.) Figure 1(c) tests large-scale inference efficiency by comparing against common acceleration techniques such as GraphSAGE Hamilton et al. (2017), pruned SAGE (PSAGE) Zhou et al. (2021), quantized SAGE (QSAGE) Zhao et al. (2020), and neighbor sampling (NS) Zou et al. (2019). HSelKD consistently yields the fastest inference across Arxiv and Products. Figure 1(d) shows that the graph-free

Table 7: Performance of HSelKD and baseline hypergraph/distillation models across benchmarks.

| Dataset | HGNN | HGNN$^+$ | DistillHGNN | GLNN | KRD | NOSMOG | LightHGNN$^+$ | HSelKD |
|---|---|---|---|---|---|---|---|---|
| CA-Cora | 72.14±2.54 | 72.79±1.28 | 75.14±1.52 | 72.19±3.83 | 71.75±3.53 | 68.96±7.34 | 73.77±1.55 | **75.42±2.02** |
| DBLP-Paper | 71.74±0.90 | 73.05±1.69 | 72.91±0.00 | 72.50±1.62 | 72.85±6.76 | 71.47±2.13 | 71.71±1.70 | **73.16±1.36** |
| IMDB-AW | 50.42±1.70 | **50.67±1.75** | 50.62±1.59 | 50.48±1.51 | 49.65±2.12 | 48.96±1.43 | 49.09±3.75 | 50.54±2.74 |

design of HSelKD improves memory efficiency, reducing usage by up to 64.71% on CC-Cora compared with the teacher HGNN.

Table 8 reports results on Large-Scale Hypergraphs Recipe-100k and Recipe-200k. Standard HGNNs fail to scale, producing OOM (out of memory) errors in inductive and production modes and showing very high latency in transductive mode (e.g., 408.9s on Recipe-100k). In contrast, HSelKD reduces inference time to 68.3s on the same task, and remains tractable for both inductive (276.4s) and production (290.7s) settings where HGNN is infeasible. On Recipe-200k, HSelKD similarly achieves faster inference (58.6s vs. 131.4s in transductive mode) while sustaining competitive accuracy. These results highlight that HSelKD preserves predictive quality while enabling scalability to million-scale hypergraphs where HGNNs become impractical.

Table 8: Experimental results on large-scale hypergraph datasets.

| Dataset | Setting | HGNN | | | HSelKD | | |
|---|---|---|---|---|---|---|---|
| | | Trans. | Ind. | Prod. | Trans. | Ind. | Prod. |
| | #Test | 10,063 | 90,562 | 100,625 | 10,063 | 90,562 | 100,625 |
| Recipe-100k | Inference time | 408.855 | ∞ | ∞ | 68.32 | 276.39 | 290.70 |
| | Accuracy | 43.35 | OOM | OOM | 41.10 | 41.40 | 41.37 |
| | #Test | 11957 | 227177 | 239134 | 11957 | 227177 | 239134 |
| Recipe-200k | Inference time | 131.35 ms | ∞ | ∞ | 58.64 | 493.80 | 524.01 |
| | Accuracy | 38.94 | OOM | OOM | 40.61 | 39.53 | 39.58 |

Table 9 reports inference times of HGNN versus HSelKD on synthetic hypergraphs of increasing size. In the simplest configuration, both teacher (HGNN) and student (HSelKD) use 8-dimensional input/output features and 2 layers, ensuring a controlled baseline. Speedup is defined as $\frac{\text{HGNN}}{\text{HSelKD}}$, and Improvement as the relative reduction in inference time. Even under this minimal setting, HSelKD achieves substantial gains, with up to 53.1× speedup and 98.1% reduction in cost at $N = 10^6$ nodes. These advantages are further amplified in realistic scenarios, where HSelKD's task-aware mechanism enables reusing a single full-scope teacher across multiple subtasks without retraining, unlike conventional KD pipelines that require teacher retraining for each label subset.

Table 9: Inference time of HGNN vs. HSelKD on synthetic hypergraphs of different sizes.

| Method | 2k | 4k | 8k | 12k | 20k | 30k | 40k | 100k | 1M |
|---|---|---|---|---|---|---|---|---|---|
| HGNN (s) | 4.0030 | 6.1171 | 8.3773 | 13.4492 | 16.3698 | 23.5181 | 29.7408 | 73.4851 | 1368.6538 |
| HSelKD (s) | 0.1652 | 0.2346 | 0.2353 | 0.3843 | 0.4480 | 0.5867 | 0.6816 | 1.5278 | 25.7518 |
| Speedup (×) | 24.2× | 26.1× | 35.6× | 35.0× | 36.5× | 40.1× | 43.6× | 48.1× | 53.1× |
| Improvement (%) | 95.9 | 96.2 | 97.2 | 97.1 | 97.3 | 97.5 | 97.7 | 97.9 | 98.1 |

**Ablation study.** To understand the contribution of each component in HSelKD, we conduct an ablation study. Figure 3a evaluates the effect of coupling-based alignment and hypergraph-level information. Removing either module results in noticeable accuracy drops across all benchmarks, confirming that both the coupling mechanism and the hyperedge-aware objective are essential for transferring structured teacher knowledge. Figure 3b examines the role of the contrastive alignment (CL) term. Excluding CL leads to severe performance degradation, particularly on CA-Cora, CC-Cora, and CC-Citeseer, where the student struggles to preserve discriminative structures without this guidance. Including CL consistently improves accuracy on all datasets, demonstrating its importance for enhancing the transfer of structural semantics and stabilizing training. In Figure 3c, we evaluate robustness under different levels of input feature corruption. Compared with MLP, both HGNN and HSelKD degrade more gracefully as the noise ratio increases. Importantly, HSelKD consistently stays close to HGNN and significantly above MLP, confirming that distilled structural knowledge improves noise tolerance and prevents severe collapse under corrupted inputs.

We further investigate the effect of the hyperparameter $\alpha$, which balances the contrastive alignment loss on student–teacher couplings against the OT-based and high-order objectives. Figure 4a reports accuracy on CA-Cora, CC-Cora, and CC-Citeseer, while Macro-F1, Macro-Precision, and Macro-Recall are provided in the appendix F.7. We observe a consistent trend across all datasets where performance rapidly increases when $\alpha$ rises from 0.1 to around 0.4, showing that moderate contrastive supervision substantially enhances the transfer of discriminative structural information. Figure 4b shows the influence of $\lambda$, which weights the OT divergence term. Across all datasets, accuracy remains relatively stable for $0.3 \leq \lambda \leq 0.7$. Moreover, the effect of $\beta$ is reported in Figure 4c. Moderate values yield the best trade-off, while very small $\beta$ reduces the benefit of structural supervision and very large $\beta$ can dominate the objective, slightly hurting accuracy.

Finally, Figure 4d analyzes performance with varying numbers of training samples per class. While all methods improve with more supervision, HSelKD consistently outperforms MLP. This highlights the efficiency of knowledge transfer even with limited labeled data, HSelKD inherits structural priors from HGNN, narrowing the gap to teacher-level accuracy.

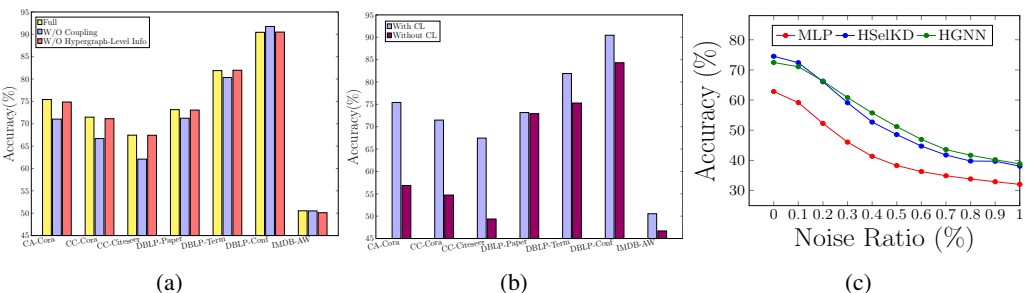

Figure 3: Effects of (a) couplings and hypergraph-level information; (b) contrastive learning; (c) feature noise.

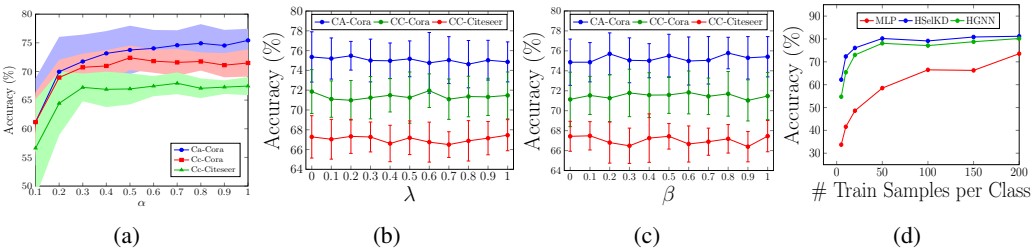

Figure 4: Sensitivity analysis of key hyperparameters.

## 5 CONCLUSION

In this work, we introduced HSelKD, a selective knowledge distillation framework that transfers the most task-relevant information from expressive yet costly HGNN teachers to lightweight MLP students. By formulating the distillation process through Inverse Optimal Transport, HSelKD achieves principled and capacity-aware alignment, reducing redundancy and mitigating negative transfer. The framework operates in two distinct modes: the task-aware mode, where the student focuses on task-relevant knowledge and specializes on meaningful subsets of labels, and the reject-aware mode, which enables the student to abstain from uncertain or out-of-distribution predictions, thereby improving robustness in deployment. Extensive experiments on diverse hypergraph and graph benchmarks demonstrate that HSelKD achieves an excellent balance between accuracy and efficiency: it consistently surpasses lightweight baselines, approaches the performance of structure-aware HGNNs, and yields significant improvements in inference latency and training cost. Overall, these results establish HSelKD as a practical, scalable, and reliable solution for deploying hypergraph-inspired models in latency-sensitive real-world applications.

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

# A    BACKGROUND AND RELATED WORK

**Hypergraph Neural Networks (HGNNs).**    Hypergraph Neural Networks (HGNNs) generalize traditional GNNs by capturing higher-order interactions among multiple nodes via hyperedges Antelmi et al. (2023); Wu & Ng (2022). Early spectral-based approaches, including HGNN Feng et al. (2019) and HpLapGCN Fu et al. (2019), leverage the hypergraph Laplacian to smooth node features across hyperedges, while HyperGCN Yadati et al. (2019) reduces hypergraphs to standard graphs, enabling the application of conventional GNN architectures. Spatial-based methods further enhance expressivity, employing vertex-hyperedge attention Bai et al. (2021), dynamic hypergraph construction Jiang et al. (2019); Yan et al. (2024); Yin et al. (2022); Hayat et al. (2024), and multi-stage message-passing schemes Dong et al. (2020); Gao et al. (2022b); Ruggeri et al. (2024) to improve information flow and adaptivity. These advances allow HGNNs to model complex, high-order relationships more effectively than traditional GNNs, with applications in social networks, recommendation systems, and biological networks.

**Knowledge Distillation from GNNs and HGNNs to Lightweight Students.**    Knowledge distillation from GNNs and HGNNs to lightweight student models has emerged as an effective strategy to combine the representational power of graph-based models with the inference efficiency of MLPs or compact GNNs. Traditional GNN-to-GNN distillation transfers knowledge from a larger teacher GNN to a smaller student GNN using techniques such as layer-wise supervision, topological structure preservation, or dynamic knowledge exchange Lassance et al. (2020); Zhang & Ma (2020); Ren et al. (2021); Joshi et al. (2021); Wu et al. (2022); Feng et al. (2022); Chen et al. (2020); Yang et al. (2020), but still inherits the latency of neighborhood aggregation. GNN-to-MLP distillation frameworks, including GLNN Zhang et al. (2021), CPF Yang et al. (2021) , and KRD Wu et al. (2023), address this by transferring soft labels or structural knowledge to MLPs, often incorporating neighbor-aware supervision or label propagation. Extending these ideas to hypergraphs, HIRE Liu et al. (2022) captures first- and second-order information in heterogeneous graphs, LightHGNN Feng et al. (2024) leverages reliable hyperedges for HGNN-to-MLP distillation, and a recent approach Yu et al. (2024) distill meta-path knowledge into hypergraph structures. These methods demonstrate that structural awareness combined with efficient student models enables faster inference while retaining expressive power. In contrast, our approach departs from conventional soft-label distillation by leveraging Optimal Transport to transfer high-order knowledge from HGNNs to MLPs, providing more reliable, topology-aware supervision.

**Hypergraph Neural Networks.**    HGNNs extend GNNs to capture high-order node interactions via hyperedges. Given a hypergraph $\mathcal{H} = (V, \mathcal{E}, X, Y, H)$ with incidence matrix $H \in \{0,1\}^{N \times M}$ and hyperedge weight matrix $W \in \mathbb{R}^{M \times M}$, HGNNs propagate information using the hypergraph Laplacian $X^{(l+1)} = \sigma\left(D_v^{-\frac{1}{2}} HWD_e^{-1}H^\top D_v^{-\frac{1}{2}} X^{(l)}\Theta^{(l)}\right)$, where $D_v \in \mathbb{R}^{N \times N}$ and $D_e \in \mathbb{R}^{M \times M}$ are the node- and hyperedge-degree matrices, $\Theta^{(l)}$ are learnable parameters of the $l$-th layer, and $\sigma(\cdot)$ is a nonlinear activation.

**Optimal Transport (OT) and Inverse OT.**    Optimal Transport (OT) provides a principled way to compare probability distributions by finding the minimal-cost plan to align them Raghvendra (2024); Shi (2025). Given distributions $\mu \in \mathbb{R}^n$ and $\nu \in \mathbb{R}^m$ over spaces $\mathcal{X}$ and $\mathcal{Y}$, and a cost matrix $C \in \mathbb{R}^{n \times m}$, OT seeks a coupling $P \in \Pi(\mu, \nu)$ that minimizes transport cost:

$$\min_{P \in \Pi(\mu,\nu)} \langle C, P \rangle = \sum_{i=1}^{n}\sum_{j=1}^{m} C_{ij}P_{ij}, \quad \Pi(\mu,\nu) = \{P \in \mathbb{R}_+^{n \times m} \mid P\mathbf{1}_m = \mu, \, P^\top\mathbf{1}_n = \nu\}. \quad (2)$$

To improve tractability, entropic regularization Wilson (1969) is often added, formulated as $\min_{P \in \Pi(\mu,\nu)} \langle C, P \rangle - \epsilon H(P)$, where $H(P) = -\sum_{i,j} P_{ij}(\log P_{ij} - 1)$ and $\epsilon > 0$ controls the regularization strength, enabling efficient Sinkhorn iterations Sinkhorn (1967). Inverse OT (IOT) inverts this process by learning the cost matrix $C_\theta$ from an observed transport plan $\widetilde{P}$ Stuart & Wolfram (2020); Chiu et al. (2022). It is typically formulated as a bi-level optimization:

$$\min_\theta \mathcal{L}(\widetilde{P} \parallel P_\theta), \quad P_\theta = \arg\min_{P \in \Pi(\mu,\nu)} \langle C_\theta, P \rangle - \epsilon H(P), \quad (3)$$

with $C_\theta$ parameterized by learnable parameters. Recent studies Shi et al. (2023) show that IOT can model fine-grained structural relations, making OT/IOT effective for structured knowledge distillation and alignment between teacher and student models.

**Graph Neural Networks (GNNs).** Graph Neural Networks (GNNs) learn on graph-structured data by iteratively aggregating information from neighbors. At layer $l$, a node $v_i$ updates its embedding via $m_i^{(l)} = \text{AGGREGATE}^{(l)}\big(\{h_j^{(l-1)} : v_j \in \mathcal{N}_i\}\big), \quad h_i^{(l)} = \text{UPDATE}^{(l)}\big(h_i^{(l-1)}, m_i^{(l)}\big)$, with $h_i^{(0)} = x_i$ as input features. By capturing pairwise dependencies, GNNs have been successfully applied in social networks, recommendation systems, and biological networks.

**Multi-Layer Perceptrons (MLPs).** MLPs are lightweight models that perform sequential linear transformations with non-linear activations.For node $i$ at layer $l$, $z_i^{(0)} = x_i$, $z_i^{(l)} = \text{Dropout}\big(\text{ReLU}\big(z_i^{(l-1)} W^{(l)} + b^{(l)}\big)\big)$, where $W^{(l)}$ and $b^{(l)}$ are trainable parameters. MLPs enable fast inference and are effective as student models in knowledge distillation frameworks guided by GNN or HGNN teachers.

**Contrastive Learning.** Contrastive learning has emerged as a powerful paradigm for self-supervised representation learning. The central idea is to learn embeddings by pulling together representations of positive pairs while pushing apart negatives, thereby structuring the embedding space based on relational similarity rather than explicit labels. A widely used formulation is the InfoNCE objective:

$$\mathcal{L}_{\text{con}} = -\sum_i \log \frac{\exp\big(\text{sim}(\mathbf{z}_i, \mathbf{z}_i^+)/\tau\big)}{\sum_j \exp\big(\text{sim}(\mathbf{z}_i, \mathbf{z}_j^-)/\tau\big)},$$

where $\text{sim}(\cdot, \cdot)$ denotes cosine similarity, $\tau$ is a temperature parameter, $\mathbf{z}_i$ and $\mathbf{z}_i^+$ form a positive pair, and $\mathbf{z}_j^-$ represents negatives.

## B   EXPERIMENTAL DETAILS AND EVALUATION METRICS

**Datasets.** We evaluate our approach on a collection of benchmark datasets consisting of three citation graphs and seven hypergraphs. The graph datasets are Cora, Pubmed McCallum et al. (2000), and Citeseer Giles et al. (1998). The hypergraph datasets include CA-Cora, CC-Cora, CC-Citeseer Yadati et al. (2019), DBLP-Paper, DBLP-Term, DBLP-Conf Sun et al. (2011), and IMDB-AW Fu et al. (2020). A statistical overview is presented in Table 10. The three citation graph datasets represent scientific publications as nodes, each labeled with its research topic. Node attributes are given by sparse bag-of-words vectors, and citation links form the edges. The hypergraph datasets can be grouped into three categories. *Publication-based hypergraphs* (CA-Cora, CC-Cora, CC-Citeseer) connect publications through either common authors (CA) or common citations (CC), with node labels corresponding to publication topics. *Author-based hypergraphs* (DBLP-Paper, DBLP-Term, DBLP-Conf) represent authors as nodes, labeled by research area. Hyperedges are formed through co-authorship on papers, shared terms, or publications in the same conference. Finally, in the *movie-based hypergraph* (IMDB-AW), nodes correspond to movies labeled by genre, and hyperedges capture two types of relations including movies sharing an actor (co-actor) or sharing a writer (co-writer).

**Transductive and Production Settings.** Node classification on graphs and hypergraphs is often evaluated under the transductive setting. In this setting, the node set $V$ is divided into a labeled set $V_L$ and an unlabeled set $V_U$. The labeled nodes are further split into a training set $V_L^{train}$ and a validation set $V_L^{val}$. During training, the model observes the entire graph $G$, including nodes in $V_L^{train} \cup V_L^{val} \cup V_U$, but only the labels of $V_L^{train}$ are used for supervision, while $V_L^{val}$ guides model selection. The labels of testing nodes in $V_U$ remain hidden, allowing the model to exploit structural information without access to ground truth.

To better mimic real-world deployment, we adopt a production setting that integrates transductive and inductive evaluations. The testing set $V_U$ is split into a transductive subset $V_U^{trand}$ and an inductive subset $V_U^{ind}$. Training is performed on a subgraph $G_{sub}$ induced by the observed

Table 10: Statistics of the benchmark datasets. Here, $\bar{d}_v$ and $\bar{d}_e$ denote the average node degree and average hyperedge size, respectively.

| Dataset | #Nodes | #Edges | #Features | $d_v$ | $d_e$ | #Classes |
|---|---|---|---|---|---|---|
| *Graph Datasets* | | | | | | |
| Cora | 2,708 | 7,440 | 1,433 | 4.8 | 2.0 | 7 |
| Pubmed | 19,717 | 54,944 | 500 | 5.5 | 2.0 | 3 |
| Citeseer | 3,327 | 6,590 | 3,703 | 3.7 | 2.0 | 6 |
| OGB-Products | 2,449,029 | 61,859,140 | 100 | 50.5 | 2.0 | 47 |
| OGB-Arxiv | 169,343 | 1,166,243 | 128 | 13.8 | 2.0 | 40 |
| *Hypergraph Datasets* | | | | | | |
| CA-Cora | 2,708 | 970 | 1,433 | 1.7 | 3.6 | 7 |
| CC-Cora | 2,708 | 1,483 | 1,433 | 2.1 | 2.1 | 7 |
| CC-Citeseer | 3,312 | 1,004 | 3,703 | 1.5 | 1.8 | 6 |
| DBLP-Paper | 4,057 | 5,701 | 334 | 2.3 | 1.6 | 4 |
| DBLP-Term | 4,057 | 6,089 | 334 | 28.6 | 19.1 | 4 |
| DBLP-Conf | 4,057 | 20 | 334 | 4.8 | 982.2 | 4 |
| IMDB-AW | 4,278 | 5,257 | 3,066 | 3.5 | 2.9 | 3 |
| Recipe-100k | 101,585 | 497,061 | 350 | 9.8 | 8.1 | 8 |
| Recipe-200k | 240,094 | 1,192,678 | 350 | 9.9 | 8.2 | 8 |

Table 11: Dataset statistics and splits under the transductive and production settings.

| Dataset | #Nodes | #Classes | Transductive Setting | | | Production Setting | | |
|---|---|---|---|---|---|---|---|---|
| | | | Train | Val | Test | Train | Val | Test (Trand / Ind) |
| Cora | 2,708 | 7 | 140 | 700 | 1,868 | – | – | – |
| Pubmed | 19,717 | 3 | 60 | 300 | 19,357 | – | – | – |
| Citeseer | 3,327 | 6 | 120 | 600 | 2,607 | – | – | – |
| CA-Cora | 2,708 | 7 | 140 | 700 | 1,868 | 140 | 700 | 1,494 / 373 |
| CC-Cora | 2,708 | 7 | 140 | 700 | 1,868 | 140 | 700 | 1,494 / 373 |
| CC-Citeseer | 3,312 | 6 | 120 | 600 | 2,592 | 120 | 600 | 2,073 / 518 |
| DBLP-Paper | 4,057 | 4 | 80 | 400 | 3,577 | 80 | 400 | 2,861 / 715 |
| DBLP-Term | 4,057 | 4 | 80 | 400 | 3,577 | 80 | 400 | 2,861 / 715 |
| DBLP-Conf | 4,057 | 4 | 80 | 400 | 3,577 | 80 | 400 | 2,861 / 715 |
| IMDB-AW | 4,278 | 3 | 60 | 300 | 3,918 | 60 | 300 | 3,134 / 783 |
| Recipe-100k | 101,585 | 8 | – | – | – | 160 | 800 | 80,500 / 20,125 |
| Recipe-200k | 240,094 | 8 | – | – | – | 160 | 800 | 191,307 / 47,826 |

nodes $V_{obs} = V_L^{train} \cup V_L^{val} \cup V_U^{trand}$. Model performance is then evaluated in three ways: on $V_U^{trand}$ (transductive evaluation), on $V_U^{ind}$ (inductive evaluation), and on the complete testing set $V_U = V_U^{trand} \cup V_U^{ind}$ (production evaluation). In our experiments, 20% of the testing nodes are allocated to inductive evaluation, with the remainder used for transductive testing, providing a realistic assessment of model performance on both observed and unseen nodes. The detailed dataset statistics and splits under both transductive and production settings are summarized in Table 11.

**Experimental Setup.** All models are trained using the Adam optimizer with a learning rate of $\eta = 0.01$ and for 200 epoch. Each experiment is repeated ten times with random seeds to account for stochasticity in weight initialization and data shuffling. In every run, 20 samples per class are randomly selected for training and 100 samples per class are allocated for validation, while the remaining data serve as the test set. Model evaluation is based on classification accuracy, and we adopt the checkpoint that achieves the best validation performance for reporting test results.Our framework is implemented in PyTorch using the DHG library. For the distillation settings, we employ an MLP as the student model, with a GCN serving as the teacher in the GNN-to-MLP case and an HGNN serving as the teacher in the HGNN-to-MLP case. All models are trained with the Adam optimizer using a learning rate of 0.01, weight decay of 0.0005, and a dropout rate of 0.5. For GCN, HGNN, HGNN$^+$, and UniGNN, the hidden dimension is set to 32 with 2 layers. GAT and UniGAT adopt a

hidden dimension of 8 with 4 attention heads and 2 layers. For MLPs, GLNN, KRD, LightHGNN$^+$, and HSelKD, we use a larger hidden dimension of 128 with 2 layers. Other hyper-parameters are aligned with the configurations reported in the original papers of each baseline. All experiments were performed on AWS EC2 ml.p3.2xlarge instances featuring a single NVIDIA V100 GPU, 8 virtual CPUs, and 61 GB of memory.

**Evaluation Metrics.** We evaluate the models using accuracy, as well as micro- and macro-averaged precision, recall, and F1-score. Let $TP_c$, $FP_c$, and $FN_c$ denote the number of true positives, false positives, and false negatives for class $c \in \{1, \ldots, K\}$. The metrics are defined as:

$$\text{Accuracy} = \frac{\sum_{c=1}^{C} TP_c}{\sum_{c=1}^{C}(TP_c + FP_c + FN_c)}$$

$$\text{Precision}_{micro} = \frac{\sum_{c=1}^{C} TP_c}{\sum_{c=1}^{C}(TP_c + FP_c)}$$

$$\text{Recall}_{micro} = \frac{\sum_{c=1}^{C} TP_c}{\sum_{c=1}^{C}(TP_c + FN_c)}$$

$$\text{Precision}_{macro} = \frac{1}{C} \sum_{c=1}^{C} \frac{TP_c}{TP_c + FP_c}$$

$$\text{Recall}_{macro} = \frac{1}{C} \sum_{c=1}^{C} \frac{TP_c}{TP_c + FN_c}$$

$$\text{F1}_{macro} = \frac{1}{C} \sum_{c=1}^{C} \frac{2 \cdot TP_c}{2 \cdot TP_c + FP_c + FN_c}$$

$$\text{F1}_{micro} = \frac{2 \cdot \text{Precision}_{micro} \cdot \text{Recall}_{micro}}{\text{Precision}_{micro} + \text{Recall}_{micro}}$$

Here, micro-averaging aggregates predictions across all classes before computing the metric, thereby weighting frequent classes more heavily. In contrast, macro-averaging computes the metric independently for each class and then averages them, treating all classes equally. This combination provides a balanced assessment across datasets with class imbalance.

## C  PSEUDO-CODE

To provide a clear overview of our proposed framework, we summarize the procedure of HSelKD and its two variants in pseudocode form. Below, we briefly describe each variant

The *target-aware* variant (Algorithm 1) limits knowledge transfer to a selected label subset $\mathcal{C}_{sel}$. If $\mathcal{C}_{sel} = \mathcal{C}$, this reduces to the standard full-task distillation. Otherwise, only task-relevant labels contribute to the training of the student, while labels outside $\mathcal{C}_{sel}$ are ignored. This design ensures that the lightweight MLP student retains its fast inference speed while focusing its capacity on the most relevant outputs. Unlike conventional knowledge distillation, our framework does not require retraining the teacher; the HGNN teacher is pre-trained once and only the necessary information is distilled to the student.

---

**Algorithm 1:** HSelKD (Selective Knowledge Distillation Framework for Hypergraphs)

---

**Input** : Hypergraph $\mathcal{H} = (V, \mathcal{E}, X, Y, H)$, Labeled set $D_L = (V_L, Y_L)$, Unlabeled set
$\quad\quad D_U = (V_U, Y_U)$, Number of epoch E
**Params:** Learning rate $\eta$, weights $(\alpha, \beta, \gamma)$, Teacher model $f_T$, student model $f_S$, label
$\quad\quad$ prototypes $\{g(y_k)\}_{k=1}^{K}$, selected class subset $\mathcal{C}_{\text{sel}}$, regularization $\epsilon$
**Output :** Predicted labels $\hat{Y}^U$ for unlabeled nodes, Trained student model parameters $\theta$

  1. Train the teacher HGNNs with label $Y^L$ ;
  2. Compute teacher logits: $z_i^T = f_T(x_i)$ ;
  3. Compute class prototypes $q_j = g(y_j)$ for $y_j \in \mathcal{C}_{\text{sel}}$ ;
  4. Build cost matrix $C_T[i, j] = -z_i^T \cdot q_j$ ;
  5. iter = 0 ;
  6. **while iter** $<$ **E do** ;
  7. $\quad\quad$ Compute $z_i^T = f_T(x_i)$ and $z_i^S = f_S^\theta(x_i)$ ;
  8. $\quad\quad$ Compute student logits: $z_i^S = f_S(x_i)$ ;
  9. $\quad\quad$ Compute cost matrix $\tilde{C}_S[i, j] = -z_i^S \cdot q_j$ ;
  10. $\quad\quad$ Solve (entropic) OT: $P_T \leftarrow \arg\min_P \langle C_T, P \rangle - \varepsilon H(P)$, $P_S^\theta$ similarly;;
  11. $\quad\quad$ Aggregate to hyperedges: $p_e^t \leftarrow \frac{1}{d_e}\sum_{v \in e}(P_T)_{v,:}, p_e^s \leftarrow \frac{1}{d_e}\sum_{v \in e}(P_S)_{v,:}$;
  12. $\quad\quad$ Compute $\mathcal{L} = \alpha \mathcal{L}_{Con}(P_S^\theta, P_T) + \beta\, KL(P_S^\theta \| P_T) + \gamma \frac{1}{|\mathcal{E}|}\sum_e JSD(p_e^s \| p_e^t)$ ;
  13. $\quad\quad$ Update $\theta \leftarrow \theta - \eta \nabla_\theta \mathcal{L}$ ;
  14. **end while** ;
  15. **return** Predicted labels $\hat{Y}^U$ for unlabeled nodes and Trained student model $f_S$ with
$\quad\quad$ parameters $\theta$ ;

---

The *reject-aware* variant (Algorithm 2) further extends this idea by enlarging the label space with an auxiliary reject class. Using a partial optimal transport formulation, the student can map uncertain or out-of-distribution instances to this reject option instead of forcing misclassification. This mechanism enhances robustness in open-world environments, where the student is expected to handle unseen or noisy inputs, while still preserving the fast inference advantage of an MLP-based model.

---

**Algorithm 2:** Reject-Aware HSelKD

---

**Input:** Teacher $f_T$, Student $f_S^\theta$, label subset $\mathcal{C}_{sel}$, reject class $r$, prototypes $g_T, g_S$, hyperedges
$\quad\quad \mathcal{E}$
**Output:** Updated student parameters $\theta$

  1. Compute teacher logits: $\text{logit}_T \leftarrow f_T(X)$;
  2. Compute student logits: $\text{logit}_S \leftarrow f_S^\theta(X)$;
  3. Extend label set with reject: $\mathcal{C}_{sel+} = \mathcal{C}_{sel} \cup \{r\}$;
  4. Build cost matrices $C_T, C_S$ using $\mathcal{C}_{sel+}$;
  5. Solve partial OT (POT): $P_T \leftarrow POT(C_T)$, $P_S^\theta \leftarrow POT(C_S)$;
  6. Aggregate hyperedge couplings $p_e^t, p_e^s$;
  7. Compute loss $\mathcal{L} = \alpha \mathcal{L}_{Con} + \beta KL + \gamma JSD$;
  8. Update $\theta \leftarrow \theta - \eta \nabla_\theta \mathcal{L}$;

---

# D   COMPUTATIONAL COMPLEXITY ANALYSIS

We analyze the per-epoch training and inference complexity of HSelKD and compare it with existing hypergraph baselines. Let $N$ be the number of nodes, $M$ the number of hyperedges, $K$ the total vertex–hyperedge incidences, $C$ the number of classes, $F$ the hidden dimension, $L$ the depth of the student MLP, $I$ the number of Sinkhorn iterations, and $N_s$ the number of masked nodes used in sub-task distillation. For HSelKD, the student forward/backward pass costs $O(LNF^2)$. The Sinkhorn coupling on the masked subset adds $O(IN_sC^2)$. Contrastive alignment introduces $O(N_s^2 C)$ if computed on the full masked set, though in practice mini-batching reduces this to $O(B^2C)$ per batch. The high-order constraint requires vertex-to-hyperedge aggregation, costing $O((K + M)C)$. Thus, the overall training complexity is

$$O\big(LNF^2 + IN_sC^2 + N_s^2C + (K + M)C\big),$$

while inference remains MLP-level at $O(LNF^2)$ since the student is graph-free.

**Comparison.** A plain MLP is the most efficient, requiring only $O(LNF^2)$ for both training and inference, but it ignores hypergraph structure and yields weak performance. HGNN incorporates high-order information at the cost of $O(LN^2F + LNF^2)$, which is often prohibitive on large graphs. LightHGNN reduces this cost to $O(LNF^2)$, matching MLP efficiency but still relying on additional teacher supervision for competitive accuracy. LightHGNN$^+$ further integrates vertex-to-hyperedge label propagation, adding $O(NMC)$ in training while inference remains $O(LNF^2)$. Finally, HSelKD introduces OT-based selective distillation, with additional terms $O(IN_sC^2 + N_s^2C + (K + M)C)$, but remains graph-free at inference with $O(LNF^2)$. In practice, HSelKD is only slightly heavier than LightHGNN during training, yet often lighter than LightHGNN$^+$ on dense hypergraphs where $M$ and $K$ are large, while preserving the same fast inference as MLP. Table 12 presents the complexity comparison of MLP, HGNN, LightHGNN, LightHGNN$^+$, and HSelKD.

Table 12: Complexity comparison of MLP, HGNN, LightHGNN, LightHGNN$^+$, and HSelKD.

| Model | Training | Inference |
|---|---|---|
| MLP | $O(LNF^2)$ | $O(LNF^2)$ |
| HGNN | $O(LN^2F + LNF^2)$ | $O(LN^2F + LNF^2)$ |
| LightHGNN | $O(LNF^2)$ | $O(LNF^2)$ |
| LightHGNN$^+$ | $O(NMC + LNF^2)$ | $O(LNF^2)$ |
| HSelKD (ours) | $O(LNF^2 + IN_sC^2 + N_s^2C + (K + M)C)$ | $O(LNF^2)$ |

**Training-Time Efficiency and Practical Overhead.** To complement the complexity analysis, we report the empirical wall-clock training times (in seconds) for LightHGNN$^+$ and HSelKD across six benchmarks in Table 13. Although HSelKD introduces a moderate training-time overhead, primarily due to Sinkhorn iterations and hyperedge-consistency terms, it remains within the same order of magnitude as LightHGNN$^+$ and is practical to train on all datasets. Crucially, the student MLP forward pass is extremely lightweight, and all additional computations are fully mini-batched, keeping the overall training cost manageable.

Importantly, HSelKD provides significantly faster inference (up to $53\times$) compared to the HGNN teacher (Table 9), and does not require retraining the heavy teacher, whereas LightHGNN$^+$ must retrain the teacher for each setting. Moreover, this moderate overhead yields substantial accuracy gains: up to +4.1 accuracy (+5.5%) in the task-aware setting (Table 5), and up to +13.23 accuracy in the reject-aware setting (Table 6).

Table 13: Wall-clock training time (s) for LightHGNN$^+$ and HSelKD across six benchmarks.

| Method | CA-Cora | CC-Cora | CC-Citeseer | DBLP-Paper | DBLP-Term | DBLP-Conf |
|---|---|---|---|---|---|---|
| LightHGNN$^+$ | 462.29 | 772.52 | 457.81 | 445.21 | 449.69 | 444.48 |
| HSelKD | 755.38 | 789.79 | 650.72 | 805.63 | 827.03 | 790.51 |

# E PROOFS OF THEORETICAL RESULTS

**Proof of Lemma 1 (Differentiability).** The student's OT plan is obtained by solving the entropic OT problem

$$P_S^\theta = \arg \min_{P \in \Pi(\mu,\nu)} \langle C_S^\theta, P \rangle - \varepsilon H(P), \quad H(P) = -\sum_{ij} P_{ij} \log P_{ij}.$$

The Sinkhorn solution has the form

$$P_S^\theta = \text{diag}(u)\, e^{-C_S^\theta/\varepsilon}\, \text{diag}(v),$$

where $u, v$ are scaling vectors. Since $\exp(\cdot)$, multiplication, and normalization are smooth operations, $P_S^\theta$ is differentiable in $C_S^\theta$. By the chain rule,

$$\frac{\partial P_S^\theta}{\partial \theta} = \frac{\partial P_S^\theta}{\partial C_S^\theta} \cdot \frac{\partial C_S^\theta}{\partial \theta}.$$

The divergences (KL, JSD, contrastive loss) are differentiable in $P_S^\theta$, hence the full HSelKD objective is differentiable in $\theta$. $\qquad\square$

**Proof of Lemma 2 (Consistency).** The KL divergence is non-negative and vanishes iff its arguments coincide:

$$KL(P_S^\theta \| P_T) \geq 0, \qquad KL(P_S^\theta \| P_T) = 0 \iff P_S^\theta = P_T.$$

With sufficient student capacity, optimization minimizes the KL to zero, forcing $P_S^\theta \to P_T$. $\qquad\square$

**Proof of Theorem 1 (Task-Aware Optimality).** Restrict the label space to $\mathcal{C}_{\text{sel}} \subseteq \mathcal{C}$. Both teacher and student transport plans now lie in the feasible domain

$$\widetilde{\mathcal{U}} = \{P \in \mathbb{R}^{n \times |\mathcal{C}_{\text{sel}}|} \mid P\mathbf{1} = \mu, P^\top \mathbf{1} = \nu\}.$$

The student objective becomes

$$\mathcal{L}(\theta) = \mathcal{L}_{\text{sup}} + \alpha \mathcal{L}_{\text{con}} + \beta \, KL(\widetilde{P}_S^\theta \| \widetilde{P}_T).$$

Since KL is minimized when $\widetilde{P}_S^\theta = \widetilde{P}_T$, the optimal solution recovers the projection of teacher knowledge onto $\mathcal{C}_{\text{sel}}$. $\qquad\square$

**Proof of Theorem 2 (Reject-Aware Robustness).** In the reject-aware setting, we use partial OT constraints

$$\mathcal{U}_{\text{POT}} = \{P \geq 0 \mid P\mathbf{1} \leq \mu, \, \mathbf{1}^\top P\mathbf{1} = \gamma\}.$$

Let $r$ denote the reject class. Suppose for an out-of-scope node $x$,

$$C_S^\theta(x, r) < C_S^\theta(x, j), \quad \forall j \in \mathcal{C}_{\text{sel}}.$$

The optimal transport minimizes $\langle C_S^\theta, P\rangle$, and since assigning mass to $r$ yields strictly lower cost, all feasible mass is transported to $r$. Therefore

$$P_S(x, r) = 1, \qquad P_S(x, j) = 0 \ \ \forall j \in \mathcal{C}_{\text{sel}}.$$

Uniqueness of the entropic POT solution ensures convergence. $\qquad\square$

**Proof of Theorem 3 (Selective Distillation Soundness).** Decompose the KL divergence over the full label space $\mathcal{C}$ into the selected subset $\mathcal{C}_{\text{sel}}$ and its complement:

$$KL(P_S^\theta \| P_T) = KL(\widetilde{P}_S^\theta \| \widetilde{P}_T) + KL(P_S^\theta(\mathcal{C} \setminus \mathcal{C}_{\text{sel}}) \| P_T(\mathcal{C} \setminus \mathcal{C}_{\text{sel}})).$$

Since KL is additive over disjoint supports and always non-negative,

$$KL(\widetilde{P}_S^\theta \| \widetilde{P}_T) \leq KL(P_S^\theta \| P_T).$$

Taking the minimum over $\theta$ on both sides yields

$$\min_\theta KL(\widetilde{P}_S^\theta \| \widetilde{P}_T) \leq \min_\theta KL(P_S^\theta \| P_T).$$

Thus, restricting to $\mathcal{C}_{\text{sel}}$ reduces or maintains divergence and avoids negative transfer. $\qquad\square$

# F  ADDITIONAL EXPERIMENTS

## F.1  T-TEST ANALYSIS BETWEEN STUDENT AND TEACHER MODELS.

To statistically evaluate the performance difference between the teacher and student models, we conducted paired Student's T-tests across multiple datasets. Table 14 reports the T-statistics and corresponding p-values for three representative datasets: CA-Cora, CC-Cora, and CC-Citeseer, under both Full Task and Reject-Aware settings.

The results show that, in all cases, the T-statistics are significantly large and the p-values are far below the common significance threshold of 0.05, indicating that the performance differences between the teacher and student models are statistically significant. Notably, the Reject-Aware setting exhibits higher T-statistics, suggesting that the student model more closely approximates the teacher's predictions when rejection mechanisms are incorporated.

These findings confirm that the student model effectively captures the knowledge from the teacher while maintaining statistically significant differences in performance across the evaluated datasets.

Table 14: Paired Student's T-test results comparing the student and teacher models.

| | Task-Aware | | | Reject-Aware | | |
|---|---|---|---|---|---|---|
| | **CA-Cora** | **CC-Cora** | **CC-Citeseer** | **CA-Cora** | **CC-Cora** | **CC-Citeseer** |
| *T-statistic* | 6.69 | 7.73 | 7.73 | 18.63 | 13.99 | 34.34 |
| *p-value* | 8.88e-05 | 2.90e-05 | 2.90e-05 | 1.69e-08 | 2.05e-07 | 7.42e-11 |

## F.2 ADDITIONAL RESULTS FOR EFFECT OF FULL-SCOPE VS. MATCHED-SCOPE TEACHER.

In addition to Accuracy (reported in the main text), we also report Macro-F1 and Micro-F1 scores for the teacher-scope comparison. As summarized in Table 15, the Full-Scope Teacher generally outperforms the Matched-Scope Teacher across CA-Cora, CC-Cora, and CC-Citeseer. The improvements are particularly notable on later subtasks, where broader label supervision yields richer semantic context and enhances the student's ability to generalize.

Table 15: Micro-F1 and Macro-F1 of Full-Scope vs. Matched-Scope teachers on 2–5 label subtasks.

| Metric | Dataset | Type | Task1 | Task2 | Task3 | Task4 | Avgerage |
|---|---|---|---|---|---|---|---|
| | CA-Cora | Full-Task | **95.09±0.78** | **83.72±2.44** | **76.01±2.18** | **73.68±3.34** | **82.13** |
| | | Sub-Task | 94.60±0.79 | 82.88±1.97 | 75.46±2.41 | 73.53±1.74 | 81.62 |
| Macro-F1 | CC-Cora | Full-Task | **93.74±1.29** | 79.80±1.59 | **72.49±1.54** | **71.03±3.54** | **79.27** |
| | | Sub-Task | 92.59±1.14 | **79.91±1.27** | 71.51±2.46 | 69.91±2.70 | 78.48 |
| | CC-Citeseer | Full-Task | **92.40±1.23** | **84.96±1.58** | **78.93±1.15** | 70.20±4.64 | **81.62** |
| | | Sub-Task | 91.71±1.09 | 83.24±1.53 | 77.40±1.68 | **71.37±0.75** | 80.93 |
| | CA-Cora | Full-Task | **95.14±0.78** | **88.86±2.02** | **81.53±2.28** | **79.15±3.40** | **86.17** |
| | | Sub-Task | 94.65±0.79 | 87.84±1.58 | 81.10±2.38 | 79.13±2.08 | 85.18 |
| Micro-F1 | CC-Cora | Full-Task | **93.78±1.30** | 86.01±1.57 | **79.11±1.96** | **77.43±3.39** | **84.08** |
| | | Sub-Task | 92.65±1.15 | **86.01±1.15** | 77.71±3.31 | 75.65±3.11 | 82.51 |
| | CC-Citeseer | Full-Task | **92.48±1.22** | **85.51±1.61** | **79.33±1.18** | 71.12±3.15 | **82.11** |
| | | Sub-Task | 91.78±1.07 | 83.81±1.51 | 77.83±1.70 | **71.60±0.76** | 81.26 |

## F.3 ADDITIONAL RESULTS FOR COMPARING HSELKD WITH LIGHTHGNN$^+$ ACROSS SUBTASKS

Alongside Accuracy (main text), we also compare HSelKD and LightHGNN$^+$ using Micro-F1 and Macro-F1. As summarized in Table 16, HSelKD consistently improves upon LightHGNN$^+$, with the largest margins observed on CC-Citeseer Task 3 (+3.74 Macro-F1). Averaged over tasks, the gains amount to +0.79 Macro-F1 on CA-Cora and +1.98 Macro-F1 on CC-Citeseer, confirming that targeted distillation strengthens generalization by transferring the most relevant teacher knowledge. Similar results for CA-Cora are reported in Table 17.

Table 16: Micro-F1 and Macro-F1 comparison of HSelKD against LightHGNN$^+$ on CC-Citeseer subtasks.

| Metric | Model | Task1 | Task2 | Task3 | Task4 | Average |
|---|---|---|---|---|---|---|
| Macro-F1 | HSelKD | **92.40±1.23** | **84.96±1.58** | **78.93±1.15** | **70.20±4.64** | **81.62** |
| | LightHGNN | 91.58±1.28 | 82.53±1.72 | 75.19±1.52 | 69.25±1.15 | 79.64 |
| | delta | +0.82 (0.90%) | +2.43 (2.94%) | +3.74 (4.97%) | +0.95 (1.37%) | +1.98 (2.49%) |
| Micro-F1 | HSelKD | **92.48±1.22** | **85.51±1.61** | **79.33±1.18** | **71.12±3.15** | **82.11** |
| | LightHGNN | 91.65±1.28 | 83.16±1.75 | 75.69±1.53 | 69.72±1.25 | 80.30 |
| | delta | +0.83 (0.91%) | +2.35 (2.83%) | +3.64 (4.81%) | +1.40 (2.01%) | +2.06 (2.57%) |

## F.4 ADDITIONAL RESULTS FOR REJECT-AWARE F1 SCORES

In addition to Accuracy reported in the main text, we also evaluate HSelKD and standard KD under reject-aware settings using Micro-F1 and Macro-F1. As summarized in Table 18, HSelKD shows

Table 17: Performance comparison of HSelKD against LightHGNN$^+$ on CA-Cora subtasks. We report Accuracy, Macro-F1, and Micro-F1 along with absolute and relative deltas. HSelKD consistently improves over LightHGNN$^+$, particularly on later subtasks where class imbalance and boundary ambiguity are stronger.

| Metric | Model | Task1 | Task2 | Task3 | Task4 | Average |
|--------|-------|-------|-------|-------|-------|---------|
| Accuracy | HSelKD | **95.14±0.77** | **88.86±2.02** | **81.53±2.28** | **79.15±3.40** | **86.17** |
| | LightHGNN | 93.98±1.05 | 88.11±2.89 | 79.53±4.81 | 78.13±4.14 | 84.94 |
| | delta | 1.16 (1.23%) | 0.75 (0.85%) | 2.00 (2.52%) | 1.02 (1.30%) | 1.23 (1.45%) |
| Macro-F1 | HSelKD | **95.09±0.78** | **83.72±2.44** | **76.01±2.18** | **73.68±3.34** | **82.13** |
| | LightHGNN | 93.92±1.06 | 83.55±3.77 | 74.32±5.01 | 73.54±3.85 | 81.33 |
| | delta | 1.17 (1.25%) | 0.17 (0.20%) | 1.69 (2.27%) | 0.14 (0.19%) | 0.79 (0.97%) |
| Micro-F1 | HSelKD | **95.14±0.78** | **88.86±2.02** | **81.53±2.28** | **79.15±3.40** | **86.17** |
| | LightHGNN | 93.98±1.05 | 88.12±2.89 | 79.53±4.81 | 78.13±4.14 | 84.94 |
| | delta | 1.16 (1.23%) | 0.74 (0.84%) | 2.00 (2.52%) | 1.02 (1.30%) | 1.23 (1.45%) |

substantial improvements over standard KD, with average gains of +10.53 Macro-F1 on CA-Cora and +8.62 Macro-F1 on CC-Citeseer. The benefits are most pronounced on harder subtasks, confirming that selective rejection enhances boundary sharpness and generalization in open-set scenarios.

Table 18: Reject-aware Micro-F1 and Macro-F1 comparison of HSelKD vs. extended vanilla KD on four subtasks.

| Metric | Dataset | Type | Task1 | Task2 | Task3 | Task4 | Avg |
|--------|---------|------|-------|-------|-------|-------|-----|
| Macro-F1 | CA-Cora | HSelKD | 78.56±2.12 | 74.24±1.94 | 71.77±2.44 | 71.47±2.73 | **74.51** |
| | | KD | 63.46±1.71 | 62.84±1.68 | 60.19±5.51 | 69.44±5.17 | 63.98 |
| | | delta | +15.10 | +11.40 | +11.58 | +2.03 | +10.53 |
| | CC-Citeseer | HSelKD | 77.46±1.87 | 74.59±1.46 | 66.80±1.77 | 61.59±1.35 | **70.11** |
| | | KD | 62.36±1.33 | 62.25±1.10 | 61.18±5.67 | 58.15±1.69 | 61.49 |
| | | delta | +15.10 | +12.34 | +5.62 | +3.44 | +8.62 |
| Micro-F1 | CA-Cora | HSelKD | 85.29±1.51 | 84.70±1.30 | 75.71±2.49 | 74.34±2.49 | **80.01** |
| | | KD | 82.53±5.62 | 81.56±6.18 | 62.48±1.26 | 69.43±6.18 | 73.50 |
| | | delta | +2.76 | +3.14 | +13.23 | +4.91 | +6.51 |
| | CC-Citeseer | HSelKD | 81.04±1.27 | 75.91±0.92 | 66.62±1.68 | 67.11±1.44 | **72.17** |
| | | KD | 73.48±5.61 | 67.67±6.71 | 62.04±4.34 | 63.07±2.65 | 66.57 |
| | | delta | +7.56 | +8.24 | +4.58 | +4.04 | +5.60 |

## F.5 ADDITIONAL RESULTS ON GRAPH BENCHMARKS

Table 19 reports results on the standard graph datasets (Cora, Pubmed, Citeseer), comparing HSelKD with representative baselines (MLP, GCN, GAT, GLNN). As shown, HSelKD consistently matches or surpasses the strongest baselines. On Cora and Pubmed, HSelKD slightly improves over GLNN and GCN, while on Citeseer it achieves the best overall accuracy (+0.37 points over GLNN). These findings indicate that the selective OT-based distillation mechanism of HSelKD remains effective not only in hypergraph settings but also when applied to conventional graph datasets.

Table 19: Accuracy of HSelKD and baselines (MLP, GCN, GAT, GLNN) across graph benchmarks.

| Dataset | MLP | GCN | GAT | GLNN | HSelKD |
|---------|-----|-----|-----|------|--------|
| Cora | 49.64±1.13 | 79.90±1.75 | 78.35±2.24 | 80.12±0.21 | **80.51±2.80** |
| Pubmed | 66.05±2.78 | 77.54±1.63 | 76.54±1.56 | 78.36±1.99 | **79.47±1.57** |
| Citeseer | 51.69±2.08 | 69.58±1.89 | 69.38±2.33 | 69.88±1.66 | **70.25±3.36** |

## F.6 VISUALIZATION

**Representation Quality.** To qualitatively assess the quality of the learned embeddings, we visualize the node representations using t-SNE for MLP, HGNN, LightHGNN$^+$, and our HSelKD (Figure 5).

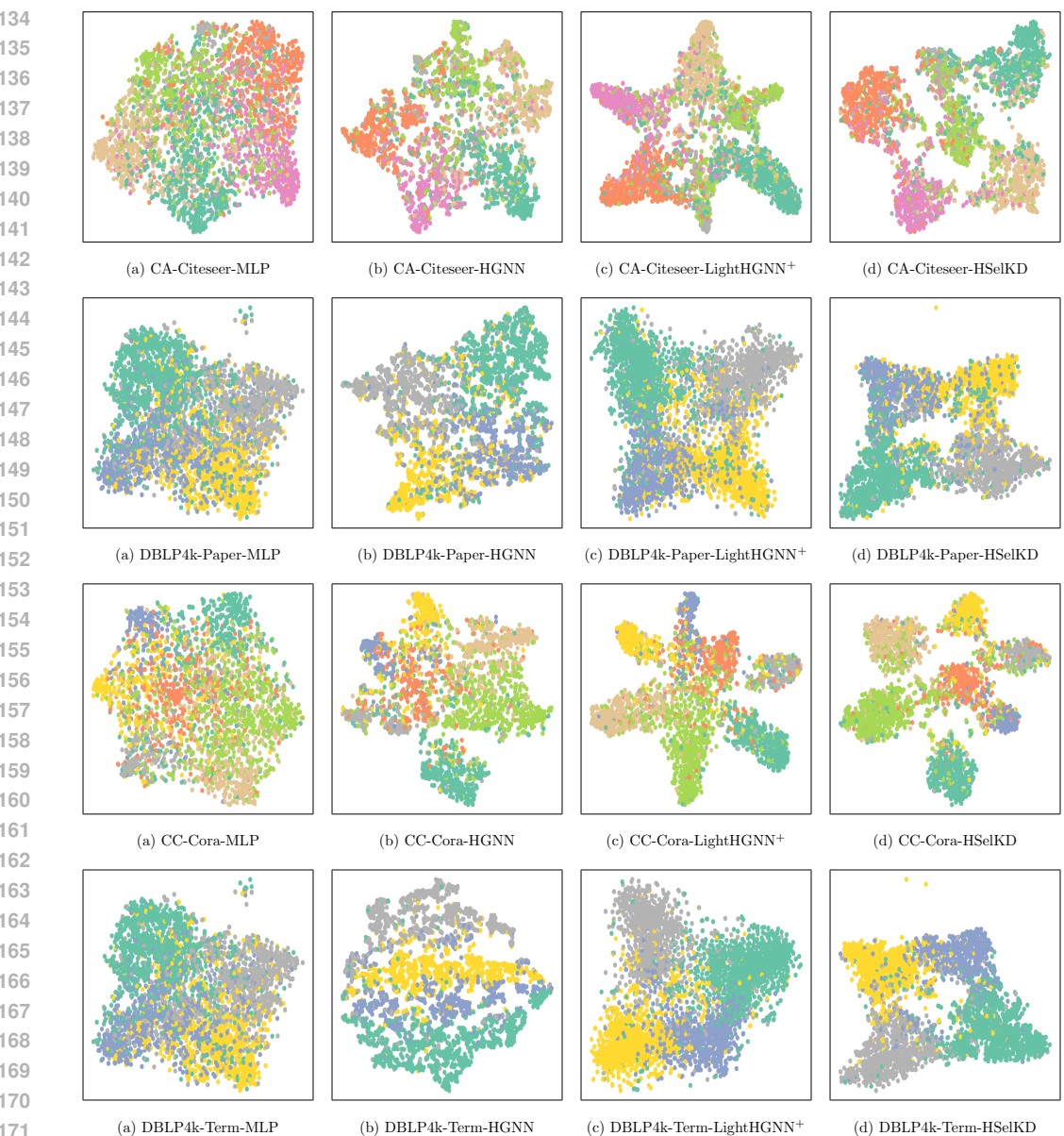

Figure 5: t-SNE visualization of the learned node representations on four benchmark hypergraph datasets(CA-Citeseer, DBLP4k-Paper, CC-Cora, and DBLP4k-Term. Each row compares representations learned by (a) MLP, (b) HGNN as teacher, (c) LightHGNN baseline, and (d) our proposed HSelKD framework. HSelKD consistently produces more discriminative and well-separated clusters, closely matching the structure captured by the teacher while preserving the efficiency of an MLP.

MLP embeddings are highly entangled, showing poor separation between classes. HGNN captures structure better but still yields overlapping regions. LightHGNN$^+$ improves cluster compactness but struggles with boundary sharpness. In contrast, HSelKD produces clearly separated and compact clusters, indicating that OT-based distillation transfers semantically meaningful and topology-aware information to the student. These findings further support the effectiveness of selective knowledge distillation in capturing high-order hypergraph semantics across diverse datasets.

**Class-Level Consistency.** To evaluate class-wise prediction behavior, we provide confusion matrix visualizations for HGNN (teacher), MLP, LightHGNN$^+$, and our proposed HSelKD across

four benchmark hypergraph datasets (Figure 6). Each matrix compares predicted versus true labels, where stronger diagonal blocks reflect higher accuracy and stronger class-level consistency.

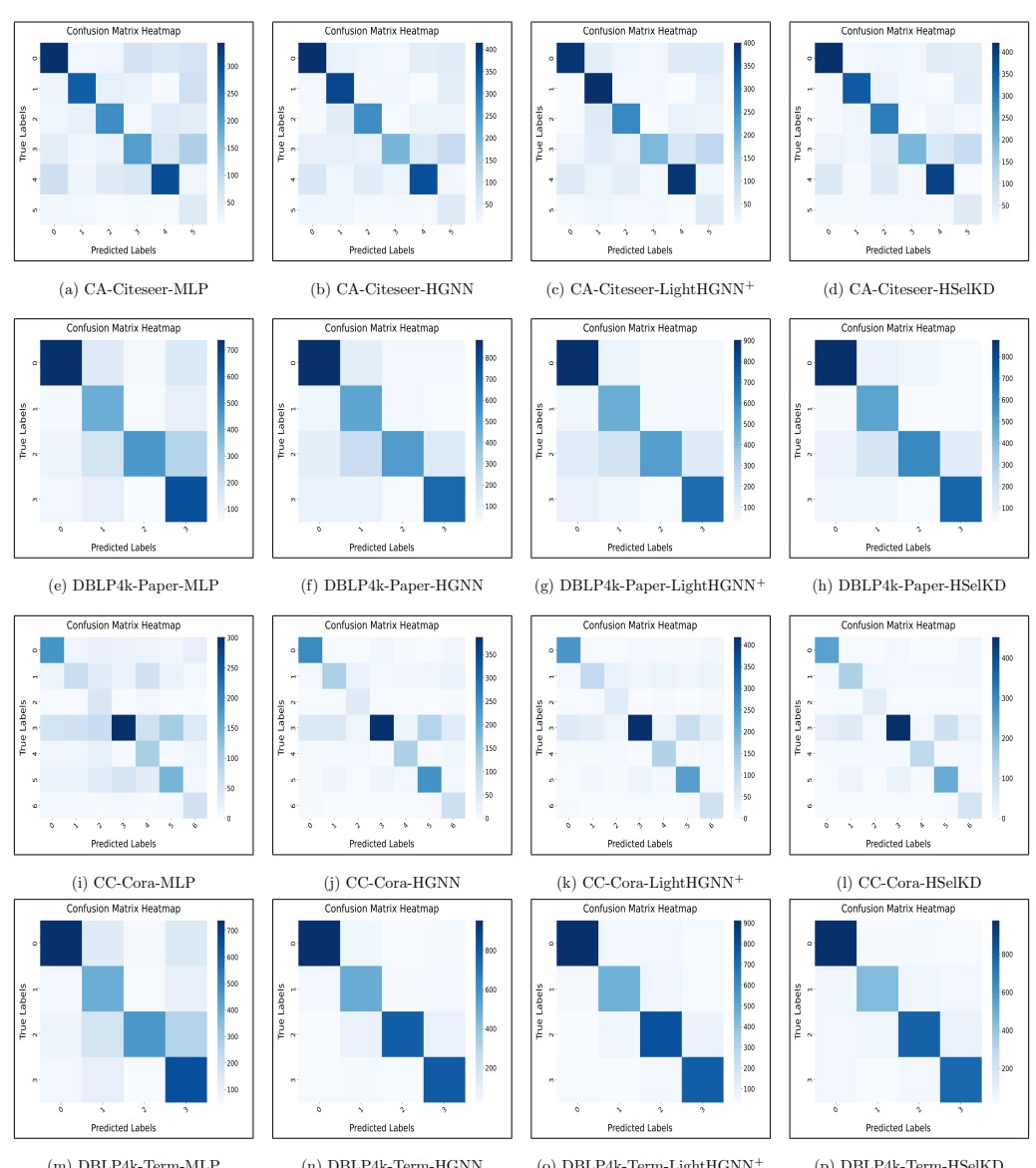

(a) CA-Citeseer-MLP  (b) CA-Citeseer-HGNN  (c) CA-Citeseer-LightHGNN$^+$  (d) CA-Citeseer-HSelKD

(e) DBLP4k-Paper-MLP  (f) DBLP4k-Paper-HGNN  (g) DBLP4k-Paper-LightHGNN$^+$  (h) DBLP4k-Paper-HSelKD

(i) CC-Cora-MLP  (j) CC-Cora-HGNN  (k) CC-Cora-LightHGNN$^+$  (l) CC-Cora-HSelKD

(m) DBLP4k-Term-MLP  (n) DBLP4k-Term-HGNN  (o) DBLP4k-Term-LightHGNN$^+$  (p) DBLP4k-Term-HSelKD

Figure 6: Confusion matrix heatmaps for node classification across four benchmark hypergraph datasets (CA-Citeseer, DBLP4k-Paper, CC-Cora, and DBLP4k-Term). Each row compares results obtained by (a) MLP, (b) HGNN as teacher, (c) LightHGNN, and (d) our proposed HSelKD framework. Darker diagonal blocks indicate stronger class separation.

As expected, the HGNN teacher produces clear diagonals, benefiting from hypergraph connectivity. The MLP baseline, lacking structural guidance, suffers from widespread off-diagonal mass, indicating severe class confusion. LightHGNN partially mitigates this through knowledge transfer, achieving more aligned diagonals but still retaining noticeable overlaps. Remarkably, HSelKD consistently yields the most dominant and well-defined diagonals across all datasets, often approaching or surpassing the teacher's consistency. These results highlight the ability of selective distillation to preserve fine-grained class-level discrimination in the student, while still enabling graph-free inference.

## F.7 SENSITIVITY ANALYSIS OF KEY HYPERPARAMETERS

We also provide Macro-F1, Macro-Precision, and Macro-Recall for investigating the effect of the hyperparameter $\alpha$. As shown in Figure 7 These metrics confirm the overall accuracy trend and further show that very small $\alpha$ values ($< 0.3$) fail to provide sufficient contrastive guidance, resulting in notable drops in Macro-F1 and Macro-Recall (up to 15% on Citeseer).

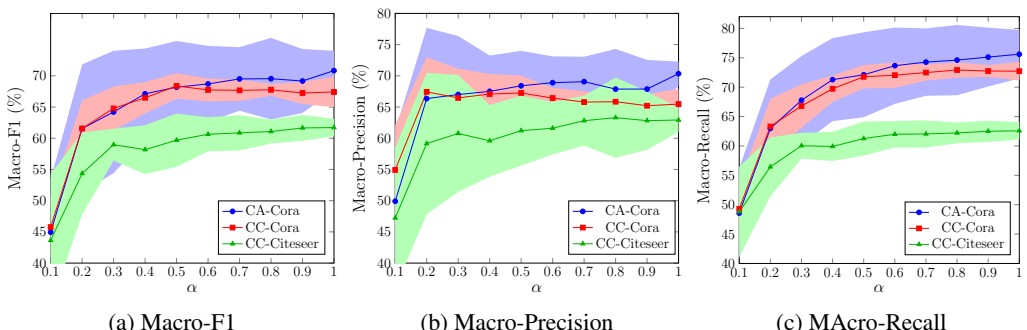

(a) Macro-F1        (b) Macro-Precision        (c) MAcro-Recall

Figure 7: Sensitivity analysis of key hyperparameters.

## F.8 ABLATION ON HYPEREDGE AGGREGATION STRATEGIES

We further evaluated alternative hyperedge aggregation schemes, including max pooling and a learnable attention-weighted variant. The results on CA-Cora, CC-Cora, and CC-Citeseer are reported in Table 20.

Table 20: Performance comparison of pooling/aggregation methods.

| Method | CA-Cora | CC-Cora | CC-Citeseer |
|---|---|---|---|
| Max pooling | $74.19 \pm 1.48$ | $71.09 \pm 1.60$ | $65.36 \pm 0.66$ |
| Attention | $74.18 \pm 2.35$ | $71.19 \pm 2.88$ | $66.70 \pm 0.79$ |
| Mean (ours) | $\mathbf{75.42 \pm 2.02}$ | $\mathbf{71.48 \pm 2.37}$ | $\mathbf{67.45 \pm 1.57}$ |

Both max pooling and attention-based aggregation yielded performance comparable to, but consistently lower than, simple averaging. Moreover, these alternatives introduced higher variance, particularly on CC-Cora and CC-Citeseer. In contrast, the mean aggregator provided the best overall accuracy and stability across datasets, owing to its permutation invariance, low variance, and computational efficiency. Consequently, we adopt averaging as the default hyperedge-level aggregation in HSelKD.

