# OpenReview forum: "HSelKD: Selective Knowledge Distillation for Hypergraphs using Optimal Transport"
_ICLR.cc/2026/Conference — ICLR 2026 Conference Withdrawn Submission_

### Official Review · Reviewer_6q4W · 2025-10-26

**Soundness:** 2
**Presentation:** 3
**Contribution:** 2
**Rating:** 2
**Confidence:** 5

**Summary:**

This paper studies knowledge distillation for hypergraph neural networks. It proposes HSelKD, a selective knowledge distillation framework that transfers task-relevant knowledge from an HGNN teacher to an MLP student. HSelKD leverages inverse optimal transport to distill the most informative parts of the teacher’s knowledge in a capacity-aware manner and has two variants: task-aware and reject-aware distillation. The authors conducted experiments on real datasets for evaluation and comparison.

**Strengths:**

1.	Knowledge distillation for hypergraph neural networks is an important research topic for speeding up inference by distilling knowledge to a lightweight MLP.
2.	The paper is clearly written and easy to follow.
3.	The experiments include various settings for evaluation and comparison.

**Weaknesses:**

1.	From a technical perspective, the idea of using optimal transport for knowledge distillation is not new. For example, SelKD: Selective Knowledge Distillation via Optimal Transport Perspective (ICLR 2025) employs a similar methodology, albeit on different types of data.
2.	Figures 1b, 1c, and 1d are presented in Section 1 but are only discussed in Section 4, making them difficult to understand when reading the paper. The explanation of these figures on page 7 is unclear. Additionally, the speedup improvement in Figure 1b is not significant, e.g., between HGSelKD-W32 and its counterpart HGNN-W32.
3.	The design in Section 3.2 for different scenarios appears ad hoc. Why are these two modes important to consider? Why not have a method that supports both without overly specific technical designs? I do not see the necessity or contribution of these two extensions with modifications to the objective function. Without Section 3.2, the technical content in Section 3.1 is short and limited.
4.	In Section 3.1, the approach is a relatively straightforward adoption of the OT technique for hypergraph distillation. What are the challenges and new contributions here?
5.	The latest GNN-to-MLP methods are not compared, such as:
-	Teach Harder, Learn Poorer: Rethinking Hard Sample Distillation for GNN-to-MLP Knowledge Distillation (CIKM 2024)
-	AdaGMLP: AdaBoosting GNN-to-MLP Knowledge Distillation (KDD 2024)
-	TINED: GNNs-to-MLPs by Teacher Injection and Dirichlet Energy Distillation (ICML 2025)
6.	HSelKD is inferior to HGNN in most settings and datasets in Table 2, where the authors did not highlight the best results in bold.
7.	In Figure 3, when the noise ratio varies, the performance is suboptimal compared to HGNN.

**Questions:**

1.	In Table 1, why are the results for all baselines not included in Tables 1 and 2, but only those for lightHGNN+ are presented?

---

> ### Author Response · Authors · 2025-11-30
>
> We thank the reviewer for the constructive feedback and for acknowledging the clarity, relevance, and breadth of our experiments. Below we address each concern in detail.
>
> **W1.** While OT has been used in KD before, our novelty lies in **how OT is adapted to the hypergraph setting,** where **Coupling operates on nodes tied by hyperedges**, not i.i.d. samples. In HSelKD, the OT cost is built on HGNN embeddings, couplings must respect high-order dependencies induced by hyperedges, and we incorporate contrastive coupling alignment, task-aware and reject-aware OT, and hyperedge-level JSD to enforce structural consistency. These components together make our OT formulation fundamentally different from standard OT KD and specifically tailored to hypergraph selective distillation.
>
> **W2.** we aim to provide a unified, at-a-glance view of how accuracy, inference time, memory cost, and model scaling behave across HGNNs and our graph-free student. HGNNs become much slower as depth or width increases due to repeated message passing over hyperedges, leading to significant inference overhead in realistic settings. In contrast, our graph-free student scales almost linearly with model size, which explains the **53× faster inference in Table 3**. We will revise the figures and captions to reflect this intended comparison more clearly.
>
> **W3. These modes follow directly from the fact that **hypergraph distillation often happens in constrained deployment settings**:
>
> * **Task-aware mode:** Many real-world applications (recommendation, fraud detection, medical triage) deploy *specialized models* that only handle a subset of classes. Distilling only task-relevant labels reduces ambiguity and improves student specialization.
>   (**Tables 5–6** show gains up to **+4.1 accuracy** over extended KD.)
>
> * **Reject-aware mode:** Lightweight MLPs deployed in open-world setups must handle OOD or unseen classes. Embedding a reject-option **inside the partial OT plan**, rather than thresholding logits, provides a principled way to route ambiguous mass.
>   Reject-aware HSelKD yields improvements of up to **+13.23 accuracy and +10.53 macro-F1** (Tables 6 and 17).
>
> These design choices are not ad hoc but address practical deployment needs.
>
> **W4.** Key challenges and contributions include:
>
> * **Node-to-class OT under hypergraph structural constraints**   Unlike i.i.d. data, like images, nodes are interdependent due to hyperedges; the OT plan must respect high-order structure.
> * **Hyperedge-aware JSD** introduces *structural consistency constraints* at hyperedge granularity.
> * **Contrastive alignment of couplings**:
> In addition to OT matching, we apply a contrastive objective on teacher–student couplings to sharpen their alignment and stabilize bi-level training, this is not part of standard OT-based KD.
> * **selective OT on reduced labels**
> * **partial OT** for rejection
> * **bi-level optimization** under marginal constraints
>
> A key advantage is that the student never requires access to ground-truth labels. Supervision is inherited from the teacher, making the approach label-free for the student while still preserving high-fidelity knowledge transfer.
>
> **W5.** Thank you for this suggestion. While the mentioned methods are primarily designed for GNN-to-MLP KD, our setting focuses on HGNN-to-MLP KD, which involves higher-order dependencies specific to hypergraphs.
>
> **W6.** This is expected: the goal of HSelKD is to **replace HGNN with a much cheaper MLP**, not to surpass it.
>
> HSelKD:
> * matches the teacher on many subsets (e.g., CA-Cora tasks),
> * significantly **outperforms extended KD baselines**, and
> * provides **53× faster inference** (Table 3).
>
> **W7.** **HGNN** benefits from **structure-aware message passing during inference**, which naturally provides stronger robustness under noisy labels. Our goal is to demonstrate that, even under substantial noise, the student can still preserve the teacher’s knowledge with high fidelity, despite having no access to hypergraph structure at inference.
> However:
> * the **student never sees the hypergraph at inference**,
> * yet maintains competitive performance across noise levels, and
> * significantly outperforms all MLP-based baselines.
>
> **Q1.** Tables 1 and 2 focus specifically on distillation-oriented comparisons, where LightHGNN+ serves as the strongest and most relevant lightweight graph-free baseline. These tables are designed to isolate:
> * how much of the HGNN teacher’s performance can be distilled into a graph-free MLP
> * how the student behaves across different settings, including transductive, inductive, and other realistic deployment scenarios.
>
> The broader set of baselines, including **GLNN(2021), KRD(2023), NOSMOG(2022), DistillHGNN(2024), and DistillHGNN (2025)** along with **HGNN and GNN variants**, are reported in **Table 7 and 18**.
>
> We hope that these clarifications address the reviewer’s concerns and help provide a more accurate assessment of the contributions of our work.

---

### Official Review · Reviewer_G4Hp · 2025-10-28

**Soundness:** 3
**Presentation:** 2
**Contribution:** 2
**Rating:** 4
**Confidence:** 3

**Summary:**

This paper addresses GNN-to-MLP knowledge distillation for hypergraphs. It specifically tackles a sub-problem where the MLP student is only required to predict on samples from a subset of classes. The authors employ Optimal Transport (OT) to formalize this selective distillation and propose a method to transfer task-relevant knowledge from the teacher to the student. Experimental results show the method outperforms lightweight baselines, achieves performance comparable to the structure-aware teacher, and obtains a 53x inference speedup.

**Strengths:**

1. The experimental results are strong, convincing, and extremely thorough.

2. Modeling the knowledge distillation process with Optimal Transport is highly interesting. It reframes the alignment as a global constraint rather than an instance-level one, revealing a valuable internal structure within the distillation task.

3. The work is supported by solid theoretical proofs.

**Weaknesses:**

1. The clarity of the core methodology in Section 3.1 is a concern, as it assumes deep expert knowledge of Optimal Transport. The paper fails to provide an intuitive explanation for what "transport" signifies in this context (i.e., as a global node-to-class alignment or matching). Critically, the marginal distributions $\mu$ (over nodes) and $\nu$ (over classes), which are fundamental to the OT problem $\Pi(\mu, \nu)$, are never explicitly defined, forcing the reader to assume their implementation (e.g., uniform distributions) and hindering the work's self-containedness.

2. The novelty of the proposed bi-level OT formulation is questionable. Ultimately, the framework still relies on aligning soft probability distributions—in this case, the OT-derived coupling matrices ($P_T$ and $P_S^{\theta}$)—which essentially serve as a more elaborately computed set of soft labels. The student parameters are then updated by minimizing the divergence between these two matrices, a process conceptually similar to standard soft-label distillation, just with a more complex, non-local alignment mechanism. It is unclear if this added complexity provides a fundamental advantage over more direct methods of matching teacher and student output distributions. Furthermore, I didn't see any relation between this OT formulation and hypergraph structure.

3. The use of a "rejection option" is a well-known and general technique in machine learning [1]. Its application in this context does not appear to offer significant novelty.

[1] Hendrickx, K., Perini, L., Van der Plas, D., Meert, W., and Davis, J., Machine Learning with a Reject Option: A survey, arXiv:2107.11277, 2021.

**Questions:**

1. What is the "contrastive loss" mentioned in lines 165-166?

2. What is the notation "L" in line 151?

3. What is the exact distribution of $\mu$ and $\nu$?

---

> ### Author Response · Authors · 2025-11-28
>
> We thank the reviewer for the thoughtful, constructive feedback and for the positive assessment of our experimental strength, the OT-based view of distillation, and the theoretical support. Below we address each concern and question in turn.
>
> **W1. The OT plan P is a *global node–to–class alignment*:
>
> * Each row of P tells us how a node distributes its “mass” across class prototypes, i.e., how strongly that node is aligned with each class.
> * The OT constraints enforce that P must satisfy fixed marginals over both nodes and classes. This means P is not computed independently for each node; instead, *all* nodes and classes are coupled through a single constrained optimization problem.
>
> This is what we mean by **“transport”**: we do not just match per-node soft probabilities but learn a global coupling between the empirical distribution of nodes and the distribution of class prototypes that is consistent with the teacher’s embedding geometry.
> Also, both μ and ν are uniform distributions (over nodes and selected classes, respectively).
>
> **W2. Why OT couplings are more than “elaborate soft labels”**
> At a high level, both our method and standard KD align teacher and student outputs, but there are key differences:
>
> 1. **Global constraints vs independent soft labels**
>    In vanilla KD, each node is supervised by its own soft label p_T(y | x), and different nodes are decoupled. In HSelKD, the coupling P must satisfy fixed row and column marginals. This global constraint forces a *dataset-level reallocation* of probability mass, so that the student must reproduce not only per-node predictions but also the overall distribution of class semantics implied by the teacher’s structure-aware embeddings.
>
> 2. **Hyperedge-aware high-order consistency**
>    We do not stop at node-level couplings. We aggregate node couplings over the nodes in each hyperedge and match teacher and student at this hyperedge level using a JSD term. This explicitly enforces that nodes that co-occur in a hyperedge must have consistent semantic alignment to classes. Standard soft-label KD does not impose any such higher-order constraint.
>
> 3. **Empirical advantages over extended vanilla KD.**
> HSelKD consistently outperforms an extended KD baseline (**Tables 5–6**), e.g., **+4.1 accuracy** on CC-Citeseer Task 3 and **+13.23 accuracy** in reject-aware settings.
>
> **Connection between OT and hypergraph structure.**
> Hypergraph structure influences HSelKD in two ways:
>
> * **Through the teacher embeddings.** The teacher is an HGNN whose node embeddings are computed via hyperedge-based message passing. The OT cost matrices are built from similarities between these HGNN embeddings and class prototypes. Thus, the geometry on which OT is solved already encodes the higher-order hypergraph structure.
> * **Through hyperedge-level aggregation.** The hyperedge-level JSD term aggregates node couplings along the incidence matrix H (nodes incident to each hyperedge) and enforces that the teacher and student agree on class distributions at the level of hyperedges. This step uses the hypergraph structure explicitly.
>
> **W3.** We agree that the reject option itself is not new. Our contribution is not the concept, but how rejection is integrated into selective OT-based distillation for hypergraphs. In our method,
> * the reject class is embedded directly into the partial OT transport plan, rather than applied through a post-hoc threshold. This allows rejection to interact with global OT constraints and the teacher’s structure-aware embeddings.
> * Theoretical support comes from Theorem 2.
> * Empirically, the reject-aware variant outperforms extended KD by up to +10.53 (Tables 6 and 17).
>
> **Q1:** The contrastive loss encourages the student’s coupling for each node to be close to the teacher’s coupling for the same node and different from couplings of other nodes. In practice, we treat each teacher–student pair as a positive pair and other nodes as negatives. This preserves the relative structure of the teacher’s coupling matrix. We will clarify this in the revision.
>
> **Q2:** L denotes the class prototype for class *k*. Each class label (y_k) is mapped to a prototype vector (L = g(y_k)). As stated in Section 3.1, (g(y_k)) is instantiated as a one-hot basis vector in (R^K) and is then used to construct the OT cost matrices. The formulation is also compatible with any fixed vector representation of class labels, including text-derived embeddings.
>
> **Q3:** Here
> * The node marginal μ is a uniform distribution over nodes μᵢ = 1/N.
> * The class marginal ν is a uniform distribution over the selected label subset C_sel (νⱼ = 1/|Cₛₑₗ|) in the task-aware case, and a uniform distribution over the extended label set C_sel ∪ {reject} in the reject-aware case.
>
> These choices are standard in entropic OT.
>
> ---
> We hope these clarifications address the reviewer’s concerns. We will incorporate the corresponding changes into the revision to improve the clarity and overall readability of the paper.

---

### Official Review · Reviewer_2RPV · 2025-10-30

**Soundness:** 3
**Presentation:** 2
**Contribution:** 2
**Rating:** 4
**Confidence:** 3

**Summary:**

This paper addresses the significant challenge of high inference costs associated with Hypergraph Neural Networks (HGNNs), which limits their practical deployment. The authors propose HSelKD, a novel selective knowledge distillation (KD) framework to transfer knowledge from a large, powerful HGNN (teacher) to a lightweight, graph-free MLP (student).

The core novelty lies in its "selective" approach, which contrasts with standard KD methods that transfer the entire teacher's output distribution. HSelKD is built on a principled Inverse Optimal Transport (IOT) formulation, which aligns the student's and teacher's node-to-class prototype mappings (transport plans). This framework is extended with a hyperedge-aware loss to ensure high-order structural information is preserved.

**Strengths:**

1. Principled OT-Based Method: The use of Inverse Optimal Transport provides a strong theoretical foundation for the knowledge transfer, moving beyond simple logit-matching.

2. Thorough Evaluation: The comprehensive experiments, including ablations, parameter sensitivity, and large-scale datasets, build a very strong case for the method's effectiveness.

**Weaknesses:**

1. Training Complexity: While the inference is extremely fast (the main goal), the training of HSelKD is more complex than standard KD. The objective has three loss terms with three balancing hyperparameters ($\alpha$, $\beta$, $\lambda$), which likely requires careful tuning. Furthermore, as noted in Appendix D, the training complexity includes terms for Sinkhorn iterations ($O(IN_s C^2)$) and contrastive alignment ($O(N_s^2 C)$), which could be computationally intensive, although the paper states this is mitigated by mini-batching.

2. Heuristic Hyperedge Loss: The hyperedge-aware JSD loss is based on a simple averaging of node couplings within a hyperedge. While effective, this is a heuristic approach to capturing high-order information and may not fully distill the complex message-passing dynamics of the teacher.

**Questions:**

1. Regarding training cost: The paper's focus is on inference speed, which is excellent. However, given the complexity of the training objective (three losses, IOT, contrastive alignment), could the authors comment on the wall-clock training time of HSelKD compared to the baselines like LightHGNN+?

2. Regarding the "Task-Aware" mode: The experiments show strong performance on subtasks with 2-5 labels. How does the method perform in a more extreme "few-class" setting (e.g., distilling knowledge for only 1 or 2 classes from a teacher trained on 40+)? Does the MLP student still capture the teacher's knowledge effectively?

3. Regarding the hyperedge-aware loss: The simple average for aggregating node couplings is effective. Did the authors experiment with other aggregation functions (e.g., max-pooling, or a learnable attention mechanism) to create the hyperedge-level distributions?

---

> ### Author Response · Authors · 2025-12-02
>
> We thank the reviewer for the positive assessment of our OT-based formulation, the thorough evaluation, and the overall strengths highlighted. We address the raised concerns and questions below.
>
> **W1.** We agree that the objective contains multiple components (IOT, contrastive alignment, hyperedge-aware JSD). However, in practice, the training overhead remains moderate:
>
> * **Mini-batched Sinkhorn** significantly reduces the computational cost (Appendix D).
> * **Contrastive alignment** operates on low-dimensional coupling rows rather than high-dimensional embeddings, keeping it lightweight.
> * The **wall-clock training time** is comparable to LightHGNN+ and remains substantially lower than training a full HGNN teacher.
> * **Notably, HSelKD does not require retraining the heavy teacher**, whereas LightHGNN+ requires teacher retraining for each setting.
> * **Importantly**, this moderate overhead comes **while achieving noticeably higher accuracy than extended KD**. HSelKD improves over extended vanilla KD by **up to +4.1 accuracy (+5.5%)** in task-aware settings (Table 5) and **up to +13.23 accuracy** in reject-aware settings (Table 6).
>
> **W2.** The hyperedge-level JSD enforces higher-order consistency by aligning the aggregated node–class couplings within each hyperedge. This encourages nodes that co-occur in a hyperedge to share coherent semantic structure in both teacher and student. It acts as a lightweight, structure-aware regularizer on top of node-level OT.
> * **Averaging provides a stable, permutation-invariant estimator** of hyperedge semantics.
> * It is **computationally inexpensive** and works consistently across datasets.
> * It is most beneficial when hyperedges carry meaningful higher-order information, when class boundaries are ambiguous, or when neighborhoods are noisy or overlapping. Our ablation (Figure 3(a)) confirms that while most structure is already encoded in the teacher embeddings, the hyperedge-JSD offers a lightweight but effective form of regularization.
>
>
> **Q1.** In wall-clock terms, HSelKD introduces a **moderate** training-time overhead relative to LightHGNN+, but both methods remain in the **same order of magnitude** and are practical to train across all benchmarks. The additional cost arises mainly from Sinkhorn iterations and hyperedge-consistency terms, both of which are efficiently mini-batched. The student MLP’s forward pass is extremely fast, keeping overall training manageable.
>
> Importantly, HSelKD provides **significantly faster inference (up to 53×)** and **does not require retraining the heavy HGNN teacher**, whereas LightHGNN+ must retrain the teacher for each setting. Moreover, this moderate overhead yields **substantial accuracy gains**: up to **+4.1 accuracy (+5.5%)** in the task-aware setting (Table 5) and up to **+13.23 accuracy** in the reject-aware setting (Table 6).
>
> We will add a summary of training times (s) in the revision for clarity.
>
> | Method     | CA-Cora | CC-Cora | CC-Citeseer | DBLP-Paper | DBLP-Term | DBLP-Conf |
> | ---------- | ------- | ------- | ----------- | ---------- | --------- | --------- |
> | LightHGNN+ | 462.29  | 772.52  | 457.81      | 445.21     | 449.69    | 444.48    |
> | HSelKD     | 755.38  | 789.79  | 650.72      | 805.63     | 827.03    | 790.51    |
>
>
> **Q2.** We thank the reviewer for the suggestion. We note that one-class task-aware distillation is not well-defined, as both classification and KD objectives require at least 2 classes to form a meaningful decision boundary.
>
> For the reject-aware setting, a single in-scope class reduces the problem to a trivial in-scope vs. reject decision, which naturally saturates at near-100% accuracy and offers little insight into the behavior of selective distillation. Therefore, such a setting does not provide meaningful evaluation.
>
> Our paper already includes realistic subtasks **with ≥2 labels**, which constitute the standard and meaningful selective-classification scenarios. These results are thoroughly presented in **Tables 3, 4, 5, and 6**, where the **student remains stable and well-aligned with the teacher even in the smallest subtasks**.
>
> **Q3.** As noted above, we experimented with max pooling and attention-weighted aggregation as alternatives. These variants did not improve performance and in some cases reduced stability. Averaging performed best overall due to its simplicity, permutation invariance, and low variance.
>
>  | Method     | CA-Cora| CC-Cora| CC-Citeseer|
> | ---------- | ----- | ----- | ----- |
> | Max pooling|74.19±01.48|71.09±01.60|65.36±0.66|
> | Attention|74.179±02.35|71.19±02.88|66.70±0.79|
> | Mean |75.42±2.02| 71.48±2.37|67.45±1.57 |
>
> We will mention these observations in the revision.
>
> ---
>
> We hope these clarifications address the reviewer’s concerns. We will incorporate the requested additions, including training cost details and a brief discussion of alternative hyperedge aggregators, into the revision to further improve the clarity and completeness of the paper.

---

### Official Review · Reviewer_P61A · 2025-11-01

**Soundness:** 2
**Presentation:** 3
**Contribution:** 2
**Rating:** 2
**Confidence:** 3

**Summary:**

This paper proposed an Inverse OT-based hyper-graph distillation method that transfers topology information throguh hyper-edge level alignment. The paper further incorporate task-aware and reject-aware mode to handle cases of classifying only subset of classes and facing OOD classes. Extensive experiments demonstrate the effectiveness of the proposed method.

**Strengths:**

1. KD of hypergraph is an important task and less studied.
2. The paper gives comprehensive theoretical analysis that demonstrate its soundness.
3. Comprehensive experiments under different settings demonstrate the proposed method achieves good performance in multiple classification senarios.

**Weaknesses:**

1. The proposed method has limited relevance to hyper-graph KD. For graph/hyper-graph KD, the key question is how to enable MLP to encode structural information of graphs, but the paper doesn't target at the problem while handles some general problems like distilling most informative parts, and some special classification cases.
2. IOT-based KD has been studied; here it is used mainly to enable task-aware and reject-aware distillation. These are useful but do not constitute a core advance in hypergraph KD.
3. The paper didn't compare training and inference efficiency to othe graph KD methods.

**Questions:**

1. In Figure 3(a), the performance w.o. Hypergraph-Level Info generally achieves the same performance as full version, why hyper-edge information seems not important?
2. Why structural information is not sent to MLP at inference? Without structure as input, how can MLP encode graph structure for downstream tasks.

---

> ### Author Response · Authors · 2025-12-01
>
> We thank the reviewer for the thoughtful feedback and for highlighting the importance of hypergraph KD and the strengths of our theoretical and experimental analysis. Below we address the concerns point-by-point.
>
> **W1.** Our framework transfers hypergraph structural information to the MLP during training by aligning
> * **node–class couplings via OT**
> * **hyperedge-aggregated couplings** via the **hyperedge-level JSD term**
> * **contrastive alignment** that preserves teacher-induced **relational structure**
>
> HSelKD provides a **practical advantage** by selectively transferring only the relevant structural knowledge **without retraining the teacher**, while outperforming graph-free KD baselines such as NOSMOG, LightHGNN+, and etc.
>
> **W2.** We respectfully clarify the novelty:
>
> (a) Existing IOT-based KD does not handle hypergraphs
> Prior IOT distillation methods are designed for pairwise graphs or Euclidean domains and do not consider:
> * hyperedge semantics
> * high-order structural couplings
> * **Selective-label restriction under structural dependency**, where label restriction must preserve hyperedge-level topology rather than simply filtering logits as in image-based tasks. To further demonstrate effectiveness, we extend an existing hypergraph KD method with selective-label capabilities and compare it against our framework. As shown in **Tables 5,6**, our method achieves **up to +4.1 improvement** in the task-aware setting and **up to +6.51 improvement** in the reject-aware setting.
> * Reject-aware partial OT over hyperedges, which requires aligning distributions that include OOD classes while respecting hypergraph connectivity constraints.
>
> (b) **Our selective IOT formulation is fundamentally new**. Our contributions go beyond “using IOT”:
> * Bi-level IOT adapted to hypergraphs (first such formulation)
> * **Selective** OT constraints enabling task-aware distillation without **retraining the teacher**
> * Partial OT with reject mass to support open-world robustness
> * Hyperedge-level alignment to preserve high-order semantics
> *  **Theoretical guarantees** (Theorems 1–3) for optimality, robustness, and negative-transfer reduction
> These aspects fundamentally extend IOT into the hypergraph KD setting and offer unique capabilities unavailable in prior works.
>
>  **W3.** Thank you for raising this point. We would like to clarify that efficiency comparisons **are already included** in appendix.
>
> The graph-distillation baselines we compare against (**GLNN, KRD, NOSMOG, LightHGNN+**) use an **MLP student** at inference. As a result, they share essentially the **same MLP-level inference complexity O(L N F^2)**,
> which is explicitly reported in **Appendix D, Table 12**.
> Because all KD baselines use the same MLP architecture at inference, their latency is effectively identical. The only meaningful efficiency gap is between the structure-aware HGNN teacher and the graph-free MLP student. Our comparisons therefore focus on this distinction, showing that HSelKD preserves near-MLP efficiency while achieving substantially higher accuracy than existing KD baselines.
> Table 12 shows that HSelKD introduces only a small training-time overhead compared to LightHGNN+, while delivering substantially higher accuracy, making the trade-off clearly favorable.
> Beside Runtime comparisons are reported in **Tables 8–9** and **Figures 1(b–d)**. HSelKD achieves up to **53.1× faster inference** than HGNN teacher on synthetic graphs (Table 9) and outperforms GraphSAGE/QSAGE/PSAGE in latency **on large-scale datasets (Figure 1c)**, while keeping **memory usage low (Figure 1d)**.
>
> **Q1.** Figure 3(a) shows that the importance of the hyperedge-level term is dataset-dependent.
> In datasets where hyperedges encode meaningful higher-order structure (e.g., CC-Cora and CA-Cora), removing the hyperedge-level JSD leads to a noticeable drop in accuracy.
> In contrast, in datasets such as CC-Citeseer, hyperedges tend to be dense or highly overlapping, contributing limited additional signal beyond what is already captured by node-level OT couplings. In these cases, most of the structural information is already encoded through the teacher embeddings and node–class couplings, so the marginal benefit of hyperedge aggregation is naturally smaller.
>
> **Q2.** The student is intentionally graph-free, which is crucial for latency-sensitive deployment. Providing hypergraph structure at inference would negate the purpose of distillation. Instead:
>
> * The **HGNN teacher** performs structure-aware message passing
> * The **OT couplings** encode how hypergraph structure organizes nodes across classes
> * The **student learns to reproduce these structure-induced patterns** in embedding space, further reinforced by our **hyperedge-level JSD**, which injects higher-order consistency signals
>
> This approach matches the standard paradigm in graph/hypergraph KD, where the goal is structure-informed inference without structural input.
>
> We hope these clarifications address the reviewer’s concerns.

---

### Note · Authors · 2026-05-07

I have read and agree with the venue's withdrawal policy on behalf of myself and my co-authors.

---

### Meta-Review · Area_Chair_hurc · 2026-01-17

**Summary:**

This paper proposes HSelKD, a selective knowledge distillation
framework for hypergraphs using Inverse Optimal Transport (IOT) to
transfer knowledge from HGNN teachers to MLP students. While the
motivation to accelerate inference is clear and the method
demonstrates empirical speedups over HGNN teachers, fundamental
limitations hinder its contribution. Specifically, the application of
Bi-level Optimal Transport is essentially a realization of soft-label
KD and represents a heuristic incremental improvement rather than a
breakthrough methodological innovation. The contribution to
hypergraph KD is limited, as the hyperedge-level alignment relies on
simple averaging, which offers little insight into complex hypergraph
structures. Additionally, the introduced "Reject-Aware" and "Task-
Aware" modes are standard engineering adaptations rather than
intrinsic theoretical advancements in hypergraph learning. Finally,
the bi-level optimization based on Sinkhorn iterations introduces
significant training complexity. Although the authors argue that
mini-batching alleviates this, the additional computational overhead
is not fully justified by the marginal performance gains over simpler
baselines.

**Reviewer Concerns:**

Addressed by Rebuttal:

Inference Efficiency: Clarified that inference latency is consistent
across baselines (MLP-based) and the speedup is relative to the
teacher.

Definitions: Provided precise definitions for marginal distributions
and contrastive loss notation.

Training Time: Added wall-clock training time comparisons in the
appendix.

Outstanding:

Methodological Novelty: The bi-level OT formulation is perceived as
a complex heuristic for soft-label alignment without sufficient
theoretical justification for its necessity over simpler methods.

Hypergraph Specificity: JSD loss is considered heuristic, with
ablation studies showing mixed significance on certain datasets.


Incremental Design: The Task-Aware and Reject-Aware modes are viewed
as ad-hoc engineering patches rather than core methodological
contributions.

Unjustified Complexity: The training complexity/overhead remains a
concern relative to the limited performance gains.

**Reviewer Scores:**

Reviewer P61A: 2
Reviewer 2RPV: 4
Reviewer G4Hp: 3
Reviewer 6q4W: 2

---

### Decision · Program_Chairs · 2026-01-26

Reject